# A Physics-preserved Transfer Learning Method for Differential Equations

**Hao-Ran Yang**
Sun Yat-Sen University
Guangzhou, China
yanghr26@mail2.sysu.edu.cn

**Chuan-Xian Ren**[*]
Sun Yat-Sen University
Guangzhou, China
rchuanx@mail.sysu.edu.cn

## Abstract

While data-driven methods such as neural operator have achieved great success in solving differential equations (DEs), they suffer from domain shift problems caused by different learning environments (with data bias or equation changes), which can be alleviated by transfer learning (TL). However, existing TL methods adopted in DEs problems lack either generalizability in general DEs problems or physics preservation during training. In this work, we focus on a general transfer learning method that adaptively correct the domain shift and preserve physical relation within the equation. Mathematically, we characterize the data domain as product distribution and the essential problems as distribution bias and operator bias. A Physics-preserved Optimal Tensor Transport (POTT) method that simultaneously admits generalizability to common DEs and physics preservation of specific problem is proposed to adapt the data-driven model to target domain, utilizing the pushforward distribution induced by the POTT map. Extensive experiments in simulation and real-world datasets demonstrate the superior performance, generalizability and physics preservation of the proposed POTT method.

## 1 Introduction

Many scientific problems, such as climate forecasting [36, 33] and industrial design [40, 2], are modeled by differential equations (DEs). In practice, DEs problems are usually discretized and solved by numerical methods since analytic solutions are hard to obtain for most DEs. However, traditional numerical solvers struggle with expensive computation cost and poor generalization ability. Recently, dealing DEs with deep neural network has attracted extensive attention. These methods can be roughly divided into two categories: physics-driven and data-driven. Optimizing neural networks with objective constructed by exact equations, physics-driven methods such as Physics-Informed Neural Networks [28, 26] have great interpretability, but suffer from the poor generalization capability across equations and the hard requirement of exact formulation of DEs. In contrast, Data-driven methods such as neural operator [24, 22] typically take the alterable function in the equations as input data and solution function as output data. Their generalization capability are markedly improved as they can cope with a family of equations rather than one.

However, the performance of data-driven methods are highly dependent on identical assumption of training and testing environments. If the testing data comes from different distribution, model performance may degrade significantly. In practice, however, applying model to different data distributions is a common requirement, e.g. from simulation data to experiment data, cross-region model application. While it is often hard to collect sufficient data from a new data domain to train a new model, transfer learning (TL) that aims to transfer model from source domain with plenty of data

---

[*]Corresponding author.

39th Conference on Neural Information Processing Systems (NeurIPS 2025).

Table 1: Simulation datasets used for experiments. Figures in $\mathcal{D}_1$, $\mathcal{D}_2$ and $\mathcal{D}_3$ are examples of (input,output) pairs from different domains in the transfer tasks. For 1-d curve plot, the filled regions represent the areas between the curve and the x-coordinate. For 2-d surface plot, the pixel value at each image pixel corresponds to the function value at the sampling point. Brighter color indicates larger value. Definition and more details about the subdomains datasets are provided in Appendix C.1.

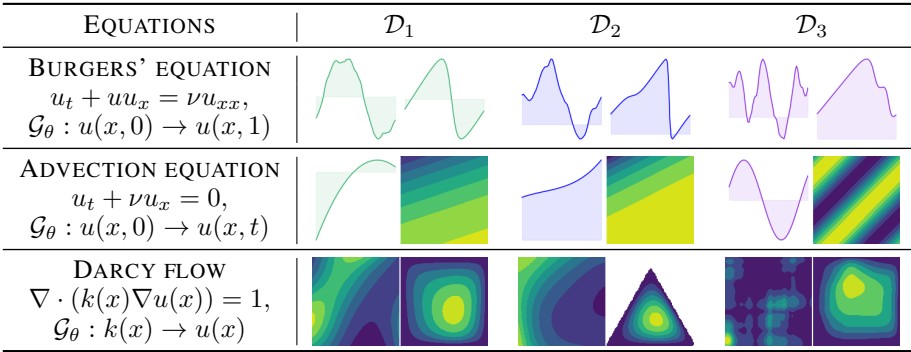

| EQUATIONS | $\mathcal{D}_1$ | $\mathcal{D}_2$ | $\mathcal{D}_3$ |
|---|---|---|---|
| BURGERS' EQUATION $u_t + uu_x = \nu u_{xx}$, $\mathcal{G}_\theta : u(x,0) \to u(x,1)$ | | | |
| ADVECTION EQUATION $u_t + \nu u_x = 0$, $\mathcal{G}_\theta : u(x,0) \to u(x,t)$ | | | |
| DARCY FLOW $\nabla \cdot (k(x)\nabla u(x)) = 1$, $\mathcal{G}_\theta : k(x) \to u(x)$ | | | |

to target domain with inadequate data, is widely adopted in real-world applications [39]. However, TL methods for DEs problems are still unexplored.

In this work, we carefully analyze the transfer learning settings in DEs problems and modelize the essential problem as distribution bias and operator bias. Given such perspective, we fully investigate the existing TL methods used in DEs problems. Technically, they can be summarized as three types. **(1) Analytic methods** [9] induce an analytic expression about model parameters to achieve adaptation with a few samples. Nevertheless, they are only feasible in very few problems with nice properties and hence not a general methodology. **(2) Finetuning** [31] the well-trained source model by target data. It is widely-used in DEs problems with domain shift due to its simplicity. Nevertheless, when the amount of available target data is limited, directly correcting operator bias by finetuning is insufficient. **(3) Domain Adaptation (DA)** [34] methods developed in other areas such as computer vision. They typically align feature distributions of source and target domains and remove the domain-specific information, so the aligned feature together with the model trained from them are domain-invariant. However, the physical relation of DEs may not necessarily be valid in the aligned feature space. Among these methods, analytic methods and finetuning directly correct the operator bias while the DA methods focus on the distribution bias correction. But all these methods have certain limitations. Therefore, a general model transfer method that can preserve the physical relation is worth exploring.

Our idea is to characterize the target domain with physics preservation and then fully correct the operator bias. Briefly speaking, we propose the Physics-preserved Optimal Tensor Transport (POTT) method to learn a physics-preserved optimal transport map between source and target domains. Then the target domain with the physical relations are characterized by the pushforward distribution, which enables a more comprehensive training for model transfer learning. Thus, the model's generalization performance on target domain can be largely improved even when only a small number of target samples are available for training. To encourage the POTT map to characterize target distribution in a physics-preserved way, we introduce a problem-specific physical regularization to the OT problem, which is derived from available physical prior of the problem. In general cases without physical prior, the regularization term is formulated as relation between marginal pushforward distribution. Our contributions are summarized as follows:

- A detailed analysis of transfer learning for DEs problems is presented, based on which we propose a feasible transfer learning paradigm that simultaneously admits generalizability to general DEs problems and physics preservation of specific problems.

- We propose POTT method to adapt the data-driven model to target domain with the pushforward distribution induced by the POTT map. A dual optimization problem is formulated to explicitly solve the optimal map. The consistency property between the solution and the ideal optimal map is presented.

- Extensive evaluation and analysis experiments on both simulation and real-world datasets are conducted. POTT shows superior performance on different types of equations with transfer tasks

of varying difficulties. Intuitive visualization analysis further supports our discussion on the physics preservation of POTT.

## 2 Preliminary

**Data-driven methods for DEs problems.** In DEs problems, data-driven methods aim to learn functional maps from data distributions. DeepONet [24] is proposed based on the universal approximation theorem [4]. Then the MIONet [16] further extends DeepONet to problems with multiple input functions and Geom-DeepONet [15] enables DeepONet to deal with parameterized 3D geometries. Differently, FNO [22] is constructed in the insight of approximating integration in the Fourier domain. Geo-FNO [21] extends FNO to arbitrary geometries by domain deformations. F-FNO [32] enhances FNO by employing factorization in the Fourier domain. Recently, transformer have also been used to construct neural operators [19, 14, 20, 35]. Although having achieved great success, these data-driven methods induce a common issue: they are highly dependent on identical assumption of training and testing environments. If the testing distribution differs, the performance of the neural operators will significantly degrade.

**Transfer learning.** Most of the transfer learning methods are proposed for Unsupervised Domain Adaptation (UDA) with classification task. They align the feature distributions of source and target domain by distribution discrepancy measurement [23], domain adversarial learning [11, 3], etc. Recent methods [5, 27, 38] further extend DA to regression settings with continuous variables. For DEs problems, analytic transfer methods [9] are presents for specific equations; finetuning [37, 31] and DA methods [12, 34] are applied in various tasks. However, there isn't a general physics-preserved transfer learning method developed for DEs problems.

**Optimal transport (OT).** OT has been quite popular in machine learning area [7, 6]. The most widely known OT problems are the Monge problem and the Kantorovich problem defined as follows:

$$M(P^s, P^t) = \inf_{T_\# P^s = P^t} \int_{\Omega^s} c(x, T(x)) \, dP^s(x) \tag{1}$$

$$K(P^s, P^t) = \inf_{\pi \in \Pi(P^s, P^t)} \int_{\Omega^s \times \Omega^t} c(x, y) \, d\pi(x, y), \tag{2}$$

where $c(x, y)$ denotes the cost of transporting $x \in \Omega^s$ to $y \in \Omega^t$, $T : \Omega^s \to \Omega^t$ denotes the transport map, $T_\# P^s$ denotes the pushforward distribution, and $\Pi(P^s, P^t)$ denotes the set of joint distributions with marginal $P^s$ and $P^t$. The Monge problem aims at a transport map $\mathcal{T}$ that minimize the total transport cost, called the Monge map. However, usually the solution of the Monge problem does not exist, so the relaxed Kantorovich poblem is more widely used. The Kantorovich problem can be solved as a linear programming problem. With an entropic regularization added, it can be fastly computed via the Sinkhorn algorithm [7]. Moreover, if $\pi^*$ takes the form $\pi^* = [id, \mathcal{T}]_\# P^s \in \Pi(P^s, P^t)$, then $\mathcal{T}$ is the Monge map. However, these optimization method don't scale well to large scale data domain and can't handle continuous probability distributions [30]. To deal with these limitations, neural OT methods are developed [30, 8, 18]. They typically train the neural network to directly approximate the OT map via constructing objective function by various OT problems [10, 13, 1].

## 3 Analysis and Motivation

### 3.1 Problem Formulation and Notations

Now we formally formulate the transfer learning settings for DEs problems. Consider two function space $\mathcal{D}_k$, $\mathcal{D}_u$ with elements $k : \Omega_k \to \mathbb{R}$, $u : \Omega_u \to \mathbb{R}$. Denote the product spaces as $\mathcal{D} = \mathcal{D}_k \times \mathcal{D}_u$ and $\Omega = \Omega_k \times \Omega_u$. Suppose there exist physical relations within the product space $\mathcal{D}$, which can be characterized as the following two forms:

$$\mathcal{F}(k, u) = 0 \text{ (Equation form)} \tag{3}$$
$$\mathcal{G}(k) = u \text{ (Operator form)} \tag{4}$$

Obviously, relation Eq. (4) is the explicit form of the implicit operator mapping determined by Eq. (3), whose existence is theoretically guaranteed under some conditions such as the implicit function theorem. Once the operator $\mathcal{G} : \mathcal{D}_k \to \mathcal{D}_u$ is solved, we can predict the desired physics quantities

$u$ for a group of $k$. However, directly solving the implicit function from Eq. (3) is often extremely difficult. In such cases, fitting $\mathcal{G}$ by neural network with collected data set $\{(k, u)\}$ provides a practical way for numerical approximation, which is exactly the goal of data-driven methods. Here we slightly abuse the notations $k$ and $u$ to represent both functions and their discretized value vectors.

An essential limitation is that the learning of the operator network $\hat{\mathcal{G}}$ depends heavily on the distribution of collected data. Let $P^s, P^t \in \mathcal{P}_\mathcal{D}$ be two product distributions supported on the source and target domain $\mathcal{D}^s, \mathcal{D}^t \subset \mathcal{D}$, respectively. Then the operator trained from them, denoted as $\hat{\mathcal{G}}^s$ and $\hat{\mathcal{G}}^t$, are in fact the approximations of $\mathcal{G}^s := \mathcal{G}|_{\mathcal{D}^s}$ and $\mathcal{G}^t := \mathcal{G}|_{\mathcal{D}^t}$. When distribution shift occurs, the operator relation also differs. So the transfer learning problem for DEs can be modelized as

$$P^s(k, u) \neq P^t(k, u), \quad \text{(Distribution bias)}$$
$$\implies \quad \mathcal{G}^s \neq \mathcal{G}^t. \quad \text{(Operator bias)} \tag{5}$$

In these situations, the model performance generally degrades if $\hat{\mathcal{G}}^s$ is directly applied to $\mathcal{D}^t$. While collecting sufficient training data is difficult in many applications scenarios, a common requirement is to transfer $\mathcal{G}^s$ to $\mathcal{D}^t$ with a few target data available. Formally, given $\hat{\mathcal{D}}^s = \{(k_i^s, u_i^s)\}_{i=1}^{n^s}$, $\hat{\mathcal{D}}^t = \{(k_j^t, u_j^t)\}_{j=1}^{n^t}$, with $n^t \ll n^s$, the task is to transfer source model $\hat{\mathcal{G}}^s$ to target domain $\mathcal{D}^t$ and approximate $\mathcal{G}^t$, i.e. to correct the operator bias. An intuitive illustration is shown in Fig. 2.

## 3.2 Methodology Analysis

Based on problem 5, existing transfer learning methods either directly correct the operator bias or indirectly correct the operator bias by aligning the feature distributions, all of which are subject to certain limitations.

**Analytic methods** directly correct the operator bias by deriving analytic expressions

$$\hat{\mathcal{G}}^t = \mathcal{H}_a(\hat{\mathcal{G}}^s, \hat{\mathcal{D}}^t), \tag{6}$$

where $\mathcal{H}_a$ denotes the ideal analytic formulation. Although exhibiting excellent interpretability, they are limited to problems with nice property and sufficient priors such as the explicit form of the equation. So they can only be applied to few problems with well-behaved equations.

**Finetuning by target domain data** directly corrects the operator bias by only further training the source model with collected target data:

$$\hat{\mathcal{G}}^t = \min_{\mathcal{G}} \mathcal{L}_{\text{task}}(\hat{\mathcal{D}}^t; \hat{\mathcal{G}}^s), \tag{7}$$

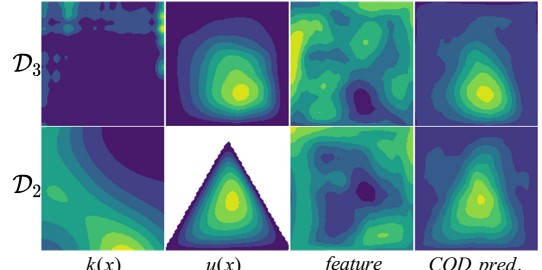

$$\mathcal{D}_3$$
$$\mathcal{D}_2$$
$$k(x) \qquad u(x) \qquad \textit{feature} \qquad \text{COD pred.}$$

Figure 1: Visualization of DA method (COD) in task $\mathcal{D}_3 \to \mathcal{D}_2$ on Darcy flow. The 1st and 2nd columns present the input $k(x)$ and output $u(x)$ sample pairs of $\mathcal{D}_3$ and $\mathcal{D}_2$. The 3rd column visualizes their feature maps from the aligned distributions, which are forced to be analogous but lacks clear structure explicit physical meaning. It is unclear whether they retain the correct physical information. The prediction shown in the 4th column verifies that the physical structures of $u$ are not fully preserved.

where $\mathcal{L}_{\text{task}}$ denotes the task-specific training loss. It does not actively and fully leverage the knowledge of source and target domain data. When the amount of available target data is limited, the predictions of target samples exhibit characteristics similar to the source samples since inadequate data is insufficient for model transfer, as discussed in Sec. 5.

**Distribution alignment methods from DA** indirectly correct the operator bias by aligning the feature distributions. These methods typically aim to learn a feature map and a corresponding feature space in which the distance between source and target feature distributions is minimized. Then a predictor trained by the source domain features can be expected to perform well on target domain features:

$$g^* = \min_g \text{dist}(g_\# P^s, g_\# P^t), \qquad \hat{\mathcal{G}}^t = \min_{\mathcal{G}} \mathcal{L}_{\text{task}}(g^*(\hat{\mathcal{D}}^s) \cup g^*(\hat{\mathcal{D}}^t); \hat{\mathcal{G}}^s), \tag{8}$$

where $\text{dist}(\cdot, \cdot)$ denotes a measurement of distribution discrepancy, g is the learned feature map, $g_\# P^s, g_\# P^t$ are the pushforward feature distributions. DA method is purely data-driven without the need of physical priors so it is a general methodology that can be used in most scenarios. However,

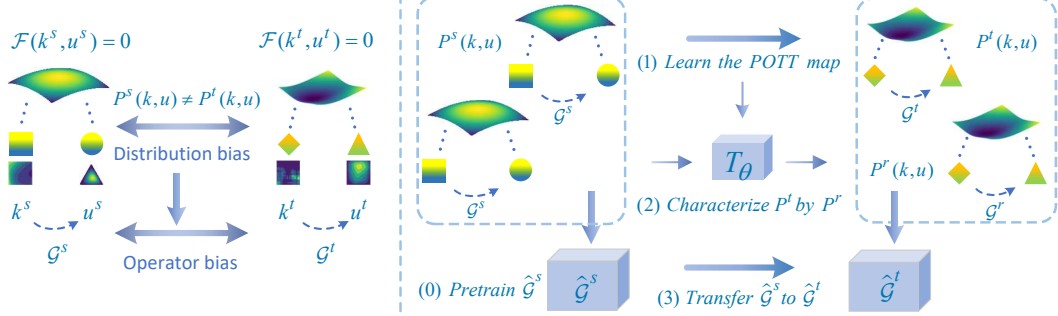

Figure 2: Illustration of POTT. **Left:** Illustration of problem formulation. The distribution bias leads to the bias of physics, i.e. the operator bias. The goal is to correct the operator bias. **Right:** POTT correct the operator bias by characterizing $\mathcal{D}^t$ in a physics-preserved way. (0) Before the model transfer process, source model $\hat{\mathcal{G}}^s$ is pretrained with sufficient source data. (1) The POTT map $T_\theta$ between source and target domain is learned. (2) The target distribution $P^t$ is characterized by the pushforward distribution $P^r$. (3) $\hat{\mathcal{G}}^s$ is transferred to $\hat{\mathcal{G}}^t$ with $\hat{\mathcal{D}}^t$ and $\hat{\mathcal{D}}^r$.

the two feature distributions are aligned by removing the domain specific knowledge so the domain invariant representations are obtained. In other words, the aligned feature distribution may lose some domain specific physical relations of both domain. As shown in Fig. 1, the features of source and target samples are analogous but confused. It is unclear whether the physical relations are preserved, which is also indicated by the output of learned $\hat{\mathcal{G}}^t$.

**Motivation of POTT.** Generally, directly correcting operator bias by finetuning only partially transfers with limited target data, while indirectly operator bias via feature distribution alignment may distort the physical relations of the DEs problem. Therefore, we propose to correct the operator bias by characterizing the target domain with physics preservation and then fully transferring $\hat{\mathcal{G}}^t$ to $\mathcal{D}^t$:

$$\hat{\mathcal{D}}^r = \mathcal{H}_c(\hat{\mathcal{D}}^s, \hat{\mathcal{D}}^t, \mathcal{R}), \quad \hat{\mathcal{G}}^t = \min_{\mathcal{G}} \mathcal{L}_{\text{task}}(\hat{\mathcal{D}}^r \cup \hat{\mathcal{D}}^t; \hat{\mathcal{G}}^s), \tag{9}$$

where $\mathcal{H}_c$ characterizes $\mathcal{D}^t$ by collected dataset $\hat{\mathcal{D}}^s$, $\hat{\mathcal{D}}^t$ and the physical regularization $\mathcal{R}$.

## 4 POTT Method

The major obstacle in Eq. (9) is to construct the $\mathcal{H}_c$. A practical way is to learn a transformation $\mathcal{T}$ between $P^s$ and $P^t$, so the target distribution $P^t$ can be characterized by the pushforward distribution $P^r = \mathcal{T}_\# P^s$, i.e. $\mathcal{H}_c(\cdot) = \mathcal{T}(\hat{\mathcal{D}}^s; \hat{\mathcal{D}}^t, \mathcal{R})$. Note that the exact corresponding relations between samples from $P^s$ and $P^t$ is unknown, so it is impractical to train $\mathcal{T}$ by traditional supervised learning. In other words, $\mathcal{T}$ shall be trained via an unpaired sample transformation paradigm. In OT theory, an optimal transport map between two distributions is the solution of an OT problem and the computation of the OT problem does not need paired samples. Therefore, a natural idea is to **model the ideal map $\mathcal{T}$ by an OT map between** $P^s$ and $P^t$. In the following sections we regard the desired map as the OT map $\mathcal{T}$, distinguishing from map $T$ that is not necessary optimal.

### 4.1 Formulation of POTT

A brief review of OT problem is provided in Sec. 2. As our purpose is to characterize $P^t$ by $P^r = T_\# P^s$, we focus on the Monge problem Eq. (1). Moreover, the physical relation in DEs can be regarded as relation between $P_k$ and $P_u$. Thus, a more reasonable perspective is to consider an OT problem between two product distributions, known as the Optimal Tensor Transport (OTT) problem. In this perspective, we propose the physics-preserved optimal tensor transport (POTT) problem:

**Definition 4.1** (POTT). Given two product distributions $P^s, P^t \in \mathcal{P}_{\mathcal{D}_k \times \mathcal{D}_u}$, define the physics-preserved optimal tensor transport (POTT) problem as:

$$\inf_{T_\# P^s = P^t} \int_{\mathcal{D}^s} c\left((k, u), T(k, u)\right) dP^s + \mathcal{R}(T), \tag{10}$$

where $T = (T_k, T_u) = (T|_{\mathcal{D}_k}, T|_{\mathcal{D}_u})$, $\mathcal{R}(T) = \mathcal{R}(T_k, T_u)$ is the physical regularization.

Specifically, when the pushforward distribution $P^r$ perfectly matches the target distribution $P^t$, the physical relation within is automatically obtained. However, it is hard to achieve in practice and $P^r$ should be regarded as an approximation or a disturbance of $P^t$. Then we expect $P^r$ to approach $P^t$ in a physics-preserved way at least, which relies on the physical regularization $\mathcal{R}$. The construction of $\mathcal{R}$ depends on the specific problem. Here we provide two basic ideas for example:

- If some physical priors of the problem are available, $\mathcal{R}$ can be derived from the priors. For example, when the weather forecasting problem is modelized as an advection partial differential equation [33], the value preservation property $\int u(x,t)dx = \text{const}, \forall t$, is a strong inductive bias. Then the $\mathcal{R}$ can be formulated as the variance of the system's value:

$$\mathcal{R}(T) = \text{var } (\int T_u u^s(x,t)dx), \tag{11}$$

  Experiment of this problem is shown in Sec. 5.

- For the general cases with no physical priors available, we formulate $\mathcal{R}$ as the physical relation between the marginal distributions of $P^r$:

$$\mathcal{R}(T) = \mathcal{R}(T_k, T_u) = m(\mathcal{G}(k^r), u^r) = m(\mathcal{G}T_k(k^s), T_u(u^s)), \tag{12}$$

  where the $m(\cdot, \cdot)$ can be any metric on $\mathcal{D}_u$. An alternative is the $L_2$ norm in the vector space. In practice, the operator $\mathcal{G}$ can be substituted by $\hat{\mathcal{G}}^s$ or $\hat{\mathcal{G}}^t$ as an approximation.

*Remark* 4.2. While both PINNs-based methods and POTT emphasize the integration of physics into the learning process, they are not directly comparable. As described in Sec. 1, PINNs-based methods rely on the explicit form of the equations to define the physics-informed loss. Consequently, they become inapplicable when the underlying equations are unknown or only partially specified. In contrast, by leveraging the optimal transport framework and incorporating physical regularization, POTT can be considered as a combination of data-driven and physics-driven method and is applicable to a broader range of problems.

## 4.2 Optimization and Analysis

Existing OTT methods [17] mainly focus on the discrete case of entropy-regularized OTT and solve the optimization problem via the Sinkhorn algorithm, which is not suitable for continuous OTT with physical regularization. Motivated by neural OT methods [30, 18], we explicitly fit $\mathcal{T}$ by a neural network and optimize it with the gradient of training loss. But Eq. (10) is a constrained optimization problem and it is challenging to satisfy the constraint during the optimization process. Therefore, we introduce the Lagrange multiplier and reformulate Eq. (10) to the unconstrained dual form.

$$\sup_f \inf_T \int_{\Omega^s} c\left((k,u), T(k,u)\right) - f(T(k,u)) + \lambda \mathcal{R}(T) \, dP^s + \int_{\Omega^t} f(k,u)dP^t. \tag{13}$$

Optimization with gradient descent tends to converge to a saddle point $(T^*, f^*)$. Following previous work [10], the consistency between $T^*$ and $\mathcal{T}$ is guaranteed.

**Theorem 4.3** (Consistency). *Suppose the dual problem Eq.* (13) *admits at least one saddle point solution, denoted as* $(T^*, f^*)$. *Let* $\mathcal{L}$ *be the objective of Eq.* (13). *Then*

- *the dual problem Eq.* (13) *equals to the Kantorovich problem with physical regularization in terms of total cost, i.e.* $\mathcal{L}(P^s, T^*, f^*) = K(P^s, P^t) + \mathcal{R}(T^*)$.

- *if* $T^*_{\#}P^s = P^t$, *then Eq.* (13) *degenerates to the dual form of the primal Monge problem Eq.* (1), $T^*$ *is a Monge map, i.e.* $\mathcal{L}(P^s, T^*, f^*) = M(P^s, P^t)$.

The proof of Thm 4.3 can be found in Appendix B. Theoretically, if the Monge map exists, i.e. $P^r = P^t$, then $P^r$ automatically admits the physics contained in $P^t$. The solution of the Monge problem is the desired OT map. However, as mentioned in Sec. 4.1, in most cases the Monge map does not exists. Therefore, the saddle point $T^*$ is not an optimal solution of the physics-regularized Monge problem. In this situation, Thm 4.3 states that Eq. (13) equals to physics-regularized Kantorovich problem in terms of total cost. Thus, the learned $T^*$ can be regarded as a compromise solution between the Monge problem and the Kantorovich problem. Importantly, the physical regularization encourages physics preservation during the training process of $T$, which is crucial for the approximation of $P^t$.

Table 2: Evaluation results of Burgers' equations.

| METHOD | $\mathcal{D}_1 \to \mathcal{D}_2$ | | $\mathcal{D}_1 \to \mathcal{D}_3$ | | $\mathcal{D}_3 \to \mathcal{D}_2$ | | AVERAGE | |
| | 50 | 100 | 50 | 100 | 50 | 100 | 50 | 100 |
|---|---|---|---|---|---|---|---|---|
| TARGET ONLY | 0.2142 | 0.1429 | 0.1173 | 0.0968 | 0.2142 | 0.1429 | 0.1819 | 0.1275 |
| SRC+TGT | 0.3960 | 0.3982 | 0.3486 | 0.3350 | 0.3145 | 0.2930 | 0.3530 | 0.3421 |
| FINETUNING | 0.2001 | 0.1191 | 0.1049 | 0.0801 | 0.1546 | 0.0938 | 0.1532 | 0.0977 |
| TL-DEEPONET | 0.1623 | 0.1182 | 0.1275 | 0.1127 | 0.1763 | 0.1436 | 0.1554 | 0.1248 |
| DARE-GRAM | 0.1727 | 0.1145 | 0.1241 | 0.1099 | 0.1752 | 0.1393 | 0.1573 | 0.1212 |
| COD | 0.1713 | 0.1225 | 0.1288 | 0.1105 | 0.1818 | 0.1525 | 0.1606 | 0.1285 |
| POTT | **0.1528** | **0.0965** | **0.0950** | **0.0705** | **0.1249** | **0.0757** | **0.1242** | **0.0809** |
| | ±0.016 | ±0.012 | ±0.007 | ±0.008 | ±0.015 | ±0.019 | ±0.013 | ±0.013 |

Table 3: Evaluation results of Darcy flow.

| TASK METHOD | $\mathcal{D}_2 \to \mathcal{D}_1$ | | $\mathcal{D}_1 \to \mathcal{D}_3$ | | $\mathcal{D}_2 \to \mathcal{D}_3$ | | AVERAGE | |
| | 50 | 100 | 50 | 100 | 50 | 100 | 50 | 100 |
|---|---|---|---|---|---|---|---|---|
| TARGET ONLY | 0.1815 | 0.1122 | 0.3925 | 0.2893 | 0.3925 | 0.2893 | 0.3222 | 0.2303 |
| SRC+TGT | 0.7113 | 0.7600 | 0.1581 | 0.1409 | 0.3535 | 0.2381 | 0.4076 | 0.3797 |
| FINETUNING | 0.1426 | 0.0869 | 0.1556 | 0.1605 | 0.4693 | 0.3553 | 0.2558 | 0.2009 |
| TL-DEEPONET | 0.1410 | 0.0805 | 0.1539 | 0.1481 | 0.4514 | 0.2842 | 0.2488 | 0.1709 |
| DARE-GRAM | 0.1395 | 0.0805 | 0.1533 | 0.1441 | 0.4509 | 0.2842 | 0.2479 | 0.1696 |
| COD | 0.1367 | 0.0794 | 0.1527 | 0.1481 | 0.4437 | 0.2836 | 0.2444 | 0.1704 |
| POTT | **0.1362** | **0.0762** | **0.1397** | **0.1404** | **0.3527** | **0.2271** | **0.2095** | **0.1479** |
| | ±0.002 | ±0.002 | ±0.009 | ±0.006 | ±0.025 | ±0.019 | ±0.012 | ±0.009 |

In practice, we parametrize the map $T$, dual multiplier $f$ and the operator $\mathcal{G}$ by neural networks $T_\theta$, $f_\phi$ and $\mathcal{G}_\eta$ with parameters denoted by $\theta$, $\phi$ and $\eta$. Function variables $k$ and $u$ are discretized into vectors. The overall objective of POTT method is

$$
\max_\phi \min_\theta \sum_{i=1}^{n^s} c((k_i^s, u_i^s), T_\theta(k_i^s, u_i^s)) - f_\phi(T_\theta(k_i^s, u_i^s)) + \lambda \mathcal{R}(T_\theta) + \sum_{j=1}^{n^t} f_\phi(k_j^t, u_j^t)
$$

$$
\min_\eta \sum_{j=1}^{n^t} \hat{\mathcal{L}}_{\text{task}}(\hat{\mathcal{G}}_\eta^t(k_j^t), u_j^t) + \beta \sum_{i=1}^{n^s} \hat{\mathcal{L}}_{task}(\hat{\mathcal{G}}_\eta^t(k_i^r), u_i^r),
$$

(14)

where $k_i^r = T_{\theta_k}(k_i^s), u_i^r = T_{\theta_u}(u_i^s)$. $\hat{\mathcal{L}}_{\text{task}}$ is the task specific loss. $\lambda$ and $\beta$ are hyper-parameters. Physical regularization term $\mathcal{R}$ depends on the available physical priors, as discussed in Sec. 4.1. A form of algorithm and the discussion of the computational cost are provided in Appendix C.

## 5 Experiment

**Benchmarks.** Experiments are conducted on both simulation and real-world datasets. For simulation datasets, three representative equations are contained. For real-world datasets, we consider the cross-region climate forecasting task. Implementation details and definition of RMSE are provided in Appendix C.

- **Simulation dataset.** Following previous works [22, 25], we take the 1-d Burgers' equation, 1-d space-time Advection equation, and 2-d Darcy Flow problem as our benchmarks. A brief introduction of these DEs problems can be found in Tab. 1. To simulate the domain shift, three different sub-domains for each equation, denoted as $\mathcal{D}_1$, $\mathcal{D}_2$ and $\mathcal{D}_3$, are generated. We generate 1000 training samples for each domain of Burgers' equation and 2000 samples for Advection equation and Darcy Flow. To fully investigate the effectiveness of transfer learning methods, we consider two scenarios that only 50 and 100 target data samples are available for model transfer.

Table 4: Evaluation results of Advection equations.

| METHOD | $\mathcal{D}_1 \to \mathcal{D}_2$ 50 | 100 | $\mathcal{D}_2 \to \mathcal{D}_1$ 50 | 100 | $\mathcal{D}_3 \to \mathcal{D}_2$ 50 | 100 | AVERAGE 50 | 100 |
|---|---|---|---|---|---|---|---|---|
| TARGET ONLY | 0.2162 | 0.1261 | 0.2585 | 0.1382 | 0.2162 | 0.1261 | 0.2303 | 0.1301 |
| SRC+TGT | 0.2347 | 0.1400 | 0.7299 | 0.4041 | 0.3969 | 0.1574 | 0.4538 | 0.2338 |
| FINETUNING | 0.0247 | 0.0143 | 0.2193 | 0.0891 | 0.1257 | 0.0723 | 0.1532 | 0.0977 |
| TL-DEEPONET | 0.0587 | 0.0127 | 0.2365 | 0.1047 | 0.1534 | 0.0685 | 0.1495 | 0.0620 |
| DARE-GRAM | 0.0572 | 0.0121 | 0.2227 | 0.0805 | 0.1687 | 0.0700 | 0.1495 | 0.0542 |
| COD | 0.0530 | 0.0120 | 0.2252 | **0.0785** | 0.1593 | 0.0644 | 0.1458 | 0.0516 |
| POTT | **0.0207** ±0.004 | **0.0112** ±0.003 | **0.1872** ±0.026 | 0.0787 ±0.017 | **0.1016** ±0.015 | **0.0613** ±0.007 | **0.1032** ±0.015 | **0.0504** ±0.009 |

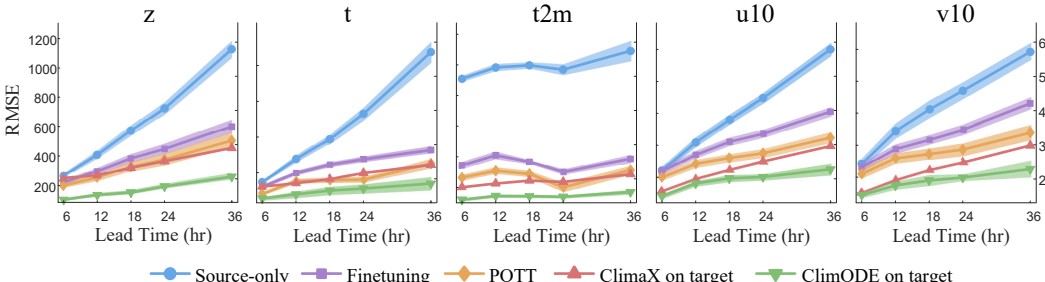

Figure 3: Comparision with (1) Source-only, source pretrained model, (2) Finetuning on target domain, (3) ClimaX, a SOTA climate forecasting model, trained with sufficient target data, (4) ClimODE, the backbone model, trained with sufficient target data. The light area reflects the standard deviation of RMSE ($\downarrow$). Details are provided in Tab. 14.

For all transfer tasks, we use 10 extra target domain samples for validation and 100 for testing. Note that the necessity of transfer learning depends on the extent of domain shift. For the transfer tasks with minor domain shift, source model can generalize well and additional transfer learning methods are unnecessary. Therefore, we only evaluate these methods in some hard tasks that need more proactive transfer. Details about data generation and how the domain shift between sub-domains is considered are provided in Tab. 8, 10,12, and Tab. 9, 11, 13 in Appendix C.1. Note that **no equation-specific information (e.g., symbolic expressions or coefficients) is provided and only** $(k, u)$ **data pairs are available during training**, which mimics real-world conditions where governing equations are unknown. Therefore, the physical regularization $\mathcal{R}$ in Eq. (12) is used for all these problems.

- **Real-world dataset.** To better assess the potential of POTT in cross-domain application of data-driven methods, we evaluate POTT in the cross-region climate forecasting task. The preprocessed $5.625°$ resolution ERA5 dataset from WeatherBench [29] is used for evaluation and the SOTA ClimODE [33] is used as backbone model. Following [33], we consider 5 quantities as label variables: ground temperature (t2m), atmospheric temperature (t), geopotential (z), and ground wind vector (u10,v10). We use climate data on North America as source domain and the global climate data as target domain. For source domain, we use ten years of training data (2006-15). For target domain, only one year (2015) of training data is available, while data of 2016 is used for validation and data of 2017-18 is used as testing data. More details are provided in Appendix C. As the ClimODE is used as backbone model, we follow the idea that modelized the climate system as an advection equation and use the physical regularization in Eq. (11) for POTT.

**Comparision methods** As discussed in Sec. 3.2, analytic methods are hard to apply in general data-driven DEs methods. So we compare POTT with finetuning and DA methods, including **Finetuning** on source pretrained model; **Src+Tgt**, training from scratch with source and target data; **TL-DeepONet** [12], a representative DA method proposed for DEs problems; **DARE-GRAM** [27] and **COD** [38], two SOTA DA methods proposed for DA regression problem with continuous variables. Since DARE-GRAM and COD are unsupervised DAR methods proposed for tasks in computer vision, we add the supervised target loss to them for fairness.

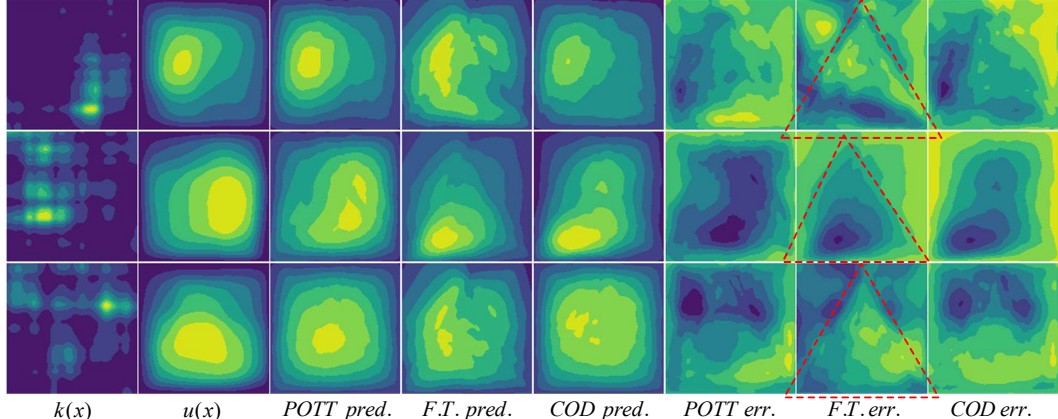

| $k(x)$ | $u(x)$ | POTT pred. | F.T. pred. | COD pred. | POTT err. | F.T. err. | COD err. |

Figure 4: Visualization of $\hat{\mathcal{G}}_t$ on Darcy Flow. The value at each image pixel represents the function value at the sampling point. Brighter colors (yellow) indicate higher values. Columns 1-2 show the input-output function pairs from the target domain. Columns 3-5 show the output of POTT, finetuning (F.T.) and COD. Columns 6-8 display the prediction errors relative to the ground truth $u(x)$.

**Evaluation results.** For simulation datasets, average rMSE of three times repeated experiments are reported. For climate forecasting, the mean and standard deviation of latitude-weighted RMSE is reported. Details about the evaluation metrics are shown in Appendix C.

- **Simulation datasets.** As shown in Tab. 3, POTT outperform finetuning and DA methods in most tasks. In tasks with severe domain shift such as $\mathcal{D}_2 \to \mathcal{D}_3$ of Darcy flow, the performances of existing methods are not satisfactory, while POTT reduces the relative error by nearly 25% (from 0.4693 to 0.3527) with only 50 target smaples available, and reduces the relative error by 36.08% (from 0.3553 to 0.2271) with 100 target samples. In simpler tasks $\mathcal{D}_2 \to \mathcal{D}_1$ and $\mathcal{D}_1 \to \mathcal{D}_3$, POTT still reduce the relative error compared to finetuning and DAR methods in every task. Results of Burgers' equation and Advection equation are provided in Tab. 2 and Tab. 4.

- **Climate forecasting.** We only compare POTT with finetuning method because the DA methods are not easily applicable for the model architecture of ClimODE. As shown in Fig. 3, directly applying ClimODE trained on North America to global forecasting results in distinct prediction error. The model performances are improved by finetuning, and POTT further reduce the prediction error, approaching the ClimODE model trained with sufficient target data. Note that the prediction error of POTT is quite closed to ClimaX model trained with sufficient target data, which is also a SOTA model in climate forecasting. Such experiment demonstrates the potential of POTT as a general model transfer method for DEs problems and real-world applications.

**Visualization analysis of prediction.** Fig. 4 illustrates the outputs and error maps for the target sample predicted by POTT, finetuning, and COD on the Darcy $\mathcal{D}_3 \to \mathcal{D}_2$ task with 100 target samples. **(1)** As shown in columns $3-5$, despite only 100 target samples are available for training, the shape and the variation trend of outputs predicted by POTT are consistent with the ground truth. In contrast, predictions of finetuning indicate that the transferred model fails to learn the right shape and variation trend. Predictions of COD are globally consistent with ground truth, but the transferred model fails to correctly predict the large value areas. **(2)** In the error maps shown in columns $6-8$, the bright areas in the error maps of POTT are the smallest among the three methods. Especially, the error maps of finetuning shown in 7th column clearly exhibit the characteristics of source domain distribution, i.e., the distinct triangular patterns, supporting the discussions in Sec. 3.2.

**Ablation analysis of physical regularization.** To investigate the effect of the physical regularization in Eq. (10), we implement an ablation analysis on the task $\mathcal{D}_1 \to \mathcal{D}_2$ of Darcy flow. As shown in Fig. 5, the outputs of OTT roughly resemble the ground truth $u_{tgt}$ in the regions of large values, but the triangular structure is not preserved, indicating the loss of certain physical

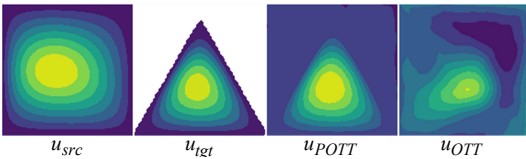

| $u_{src}$ | $u_{tgt}$ | $u_{POTT}$ | $u_{OTT}$ |

Figure 5: Visualization of POTT and OTT.

relations. In contrast, with physical regularization, the outputs of POTT exhibit consistency in both the regions of large values and the triangular structure, verifying the preservation of physics.

To more intuitively demonstrate the effect of physical regularization, we present the numerical results of $\hat{\mathcal{G}}_t$ trained using the outputs of OTT and POTT, as shown in Tab. 5. It is observed that POTT achieves lower prediction error than OTT, indicating that the outputs of POTT better characterize the target distribution in a physics-preserved manner. Furthermore, both OTT and POTT outperform vanilla finetuning.

| | Finetuning | POTT w/o $\mathcal{R}$ | POTT |
|---|---|---|---|
| $\mathcal{D}_1 \to \mathcal{D}_2$ | 0.0575 | 0.0543 | 0.0496 |

Table 5: Results of the ablation analysis.

**Extreme few-shot settings.** In extreme few-shot settings (e.g., less than 10 target samples), both finetuning and POTT indeed face significant challenges due to the risk of overfitting. From a distribution-matching perspective, fitting a meaningful target distribution with only a few samples is fundamentally ill-posed and prone to instability. This limitation applies broadly to all distribution-based methods. To empirically evaluate this, we conduct a scale-down experiment on the Darcy flow task $\mathcal{D}_2 \to \mathcal{D}_3$ under various numbers of labeled target samples $n^t$. As shown in Tab. 6,

| $n^t$ | 100 | 50 | 20 | 10 | 5 | 1 |
|---|---|---|---|---|---|---|
| Tgt. only | 0.3943 | 0.3925 | 0.4130 | **0.4897** | **0.5107** | **0.5568** |
| Finetuning | 0.3553 | 0.4693 | 0.4866 | 0.5368 | 0.5521 | 0.6282 |
| POTT | **0.2271** | **0.3527** | **0.4126** | 0.5257 | 0.5364 | 0.6359 |

Table 6: Results of scale-down experiment.

- As $n^t$ decreases, all methods degrade in performance.
- When $n^t \geqslant 20$, POTT still provides advantages over both finetuning and training from scratch.
- When $n^t \geqslant 10$, both POTT and finetuning begin to overfit, and performance starts to deteriorate significantly.
- At the extreme case of $n^t = 1$, POTT fails entirely, as the transport map becomes unreliable and may even hurt the learning of the target predictor due to inaccurate distribution matching.

These results clarify that POTT is most effective in the moderate few-shot regime, but may not be applicable when labeled target data is extremely scarce, which aligns with many real-world cases in scientific machine learning where acquiring a small number of high-quality labels is feasible.

## 6   Conclusion

In this work, we studied the domain shift issue in DEs problems. The essential problem is modeled as distribution bias and operator bias. Then we detailedly analyzed existing transfer learning methods used in DEs problems and propose a feasible POTT method that simultaneously admits generalizability to common DEs and physics preservation of specific problem. Based on the the availability of physical prior, two forms of realization of are introduced to encourage the POTT map to characterize target distribution in a physics-preserved way. A dual optimization problem and the consistency property are formulated to explicitly solve the optimal map. Extensive evaluation and analysis experiments on both simulation and real-world datasets with varying difficulties validate the effectiveness of POTT for data-driven methods in DEs problems.

**Limitations and future works.** (1) The core of POTT lies in characterizing target distribution with limited samples. However, similar to other distribution-based methods, POTT's efficacy diminishes when the quantity of available target data is extremely small, rendering its potential in one-shot or few-shot scenarios. Moreover, when abundant target data are accessible, finetuning or even training from scratch is sufficient for cross-domain application. Therefore, POTT is more valuable when the amount of target sample is small, yet not extremely small, as shown in Sec. 5 and Sec. D. (2) The learning of POTT relies on the min-max optimization, which is complex and costly for models with large-scale parameters. The resolution of the experimental data is relatively low, thereby necessitating a less complex model. However, in DEs problems and application scenarios that require high resolution, a larger model is needed to fit POTT. To address this issue, one idea is to leverage the well-developed multimodal large models to reduce the training load of the method, which will further enhance the practicality and broad applicability of this method. We believe this will contribute to the generalizability and universality of the scientific and technological achievements, such as the cross-regional climate forecasting tried in this paper.

## Acknowledgments

This work is supported in part by National Key R&D Program of China (2024YFA1011900), National Natural Science Foundation of China (Grant No. 62376291), Guangdong Basic and Applied Basic Research Foundation (2023B1515020004), Science and Technology Program of Guangzhou (2024A04J6413), and the Fundamental Research Funds for the Central Universities, Sun Yat-sen University (24xkjc013).

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

# A  Notations

The notations appear in this paper are summarized as follows:

Table 7: Notations.

| Notation | Description |
|---|---|
| $\mathcal{D} = \mathcal{D}_k \times \mathcal{D}_u = \{(k, u)\}$ | General function set |
| $\Omega = \Omega_k \times \Omega_u$ | Domain of function |
| $k = k(x)$ | function defined on $\Omega_k$ |
| $u = u(x)$ | function defined on $\Omega_u$ |
| $\mathcal{F}(x; k, u)$ | Differential Equation |
| $\mathcal{G} : \mathcal{D}_k \to \mathcal{D}_u$ | Operator map from $\mathcal{D}_k$ to $\mathcal{D}_u$ |
| $P(k, u)$ | Product probability function defined on function set $\mathcal{D}$ |
| $p(k, u)$ | Product probability density function of function set $\mathcal{D}$ |
| $\mathcal{P}_\mathcal{D}$ | Set of probability functions defined on $\mathcal{D}$ |
| $\mathcal{D}^s$ | Soure domain; a subset of $\mathcal{D}$ |
| $\mathcal{D}^t$ | Target domain; a subset of $\mathcal{D}$ |
| $\mathcal{G}^s$ | Operator relation on $\mathcal{D}^s$ |
| $\mathcal{G}^t$ | Operator relation on $\mathcal{D}^t$ |
| $P^s$ | Probability function on $\mathcal{D}^s$ |
| $P^t$ | Probability function on $\mathcal{D}^t$ |
| $T : \mathcal{D}^s \to \mathcal{D}^t$ | Function map between $\mathcal{D}^s$ and $\mathcal{D}^t$ |
| $\mathcal{T}$ | Ideal solution of optimal transport problem |
| $M(P^s, P^t)$ | Monge problem between probability $P^s$ and $P^t$ |
| $K(P^s, P^t)$ | Kantorovich problem between probability $P^s$ and $P^t$ |
| $\mathcal{R}(T)$ | Physical regularization on $T$ |
| $m(\cdot, \cdot)$ | Metric defined on $\mathcal{D}_u$ |
| $M_{phy}(P^s, P^t)$ | Monge problem with physical regularization |
| $K_{phy}(P^s, P^t)$ | Kantorovich problem with physical regularization |
| $c(\cdot, \cdot)$ | Cost function in OT problem |
| $f$ | Lagrange multiplier |
| $\mathcal{L}$ | Objective function of optimization problem |
| $(T^*, f^*)$ | Saddle point solution of dual problem |
| $\hat{\mathcal{G}}^s$ | Approximated operator on $\mathcal{D}^s$ |
| $\hat{\mathcal{G}}^t$ | Approximated operator on $\mathcal{D}^t$ |
| $\hat{T}$ | Approximation of $T$ |

we slightly abuse the notations $k$ and $u$ to represent both functions and their discretized value vectors. The superscript s or t denotes the domain. The subscript $k$ or $u$ denotes the projection of the original product space or distribution.

# B  Theory and method

## B.1  Derivation of dual formula

Given POTT problem

$$\inf_{T_\# P^s = P^t} \int_{\mathcal{D}^s} c\left((k, u), T(k, u)\right) dP^s + \mathcal{R}(T), \tag{15}$$

where $T = (T_k, T_u)$, $T_k = T|_{\mathcal{D}_k}, T_u = T|_{\mathcal{D}_u}$, we reorganize it as a constrained optimization problem:

$$\inf_T \int_{\mathcal{D}^s} c\left((k, u), T(k, u)\right) dP^s + \mathcal{R}(T) \tag{16}$$

$$s.t. T_\# P^s = P^t \tag{17}$$

Following the dual optimization theory, we introduce the Lagrange multiplier $f$ to construct the Lagrange function:

$$\mathcal{L}(T, f) = \int_{\Omega_k \times \Omega_u} c\left((k, u), T(k, u)\right) dP^s + \lambda \mathcal{R}(T) + \int_{\Omega_k \times \Omega_u} f(k, u) d(P^t - T_\# P^s). \tag{18}$$

As the physical regularization regularized the pushforward distributions $P^r = T_\# P^s$, it can be considered with the source distribution. Then it comes to

$$\mathcal{L}(T, f) = \int_{\Omega_k \times \Omega_u} c\left((k, u), T(k, u)\right) - f(T(k, u)) + \lambda \mathcal{R}(T) dP^s + \int_{\Omega_k \times \Omega_u} f(k, u) dP^t. \quad (19)$$

And the dual problem of Eq. (15) is

$$\sup_f \inf_T \ \mathcal{L}(T, f), \quad (20)$$

which is exactly Eq. (13).

## B.2   Proof of Thm. 4.3

We prove Thm. 4.3 based on previous work [10].

**Theorem B.1** (Consistency). *Suppose the dual problem Eq. (13) admits at least one saddle point solution, denoted as $(T^*, f^*)$. Let $\mathcal{L}$ be the objective of Eq. (13). Then*

- *the dual problem Eq. (13) equals to the Kantorovich problem with physical regularization in terms of total cost, i.e. $\mathcal{L}(P^s, T^*, f^*) = K(P^s, P^t) + \mathcal{R}(T^*)$.*

- *if $T^*_\# P^s = P^t$, then Eq. (13) degenerates to the dual form of the primal Monge problem Eq. (1), $T^*$ is a Monge map, i.e. $\mathcal{L}(P^s, T^*, f^*) = M(P^s, P^t)$.*

*Proof.* **(1)** Let $k^r = T_{\theta_k}(k^s), u^r = T_{\theta_u}(u^s)$, the inner optimization problem can be formulated as

$$
\begin{aligned}
&\inf_T \mathcal{L}(T, f) \\
&= \inf_T \int c\left((k^s, u^s), T(k^s, u^s)\right) - f(T(k^s, u^s)) + \lambda \mathcal{R}(T) dP^s + \int f(k^t, u^t) dP^t \\
&= -\int \sup_{(\xi^r, \zeta^r)} \{f(\xi^r, \zeta^r) - [c\left((k^s, u^s), (\xi^r, \zeta^r)\right) + \lambda \mathcal{R}(T)]\} dP^s + \int f(k^t, u^t) dP^t \\
&= \int f(k^t, u^t) dP^t - \int f^{c,-}(k^s, u^s) dP^s,
\end{aligned}
\quad (21)
$$

where

$$f^{c,-}(k^s, u^s) = \sup_{(\xi^r, \zeta^r)} \left(f(\xi^r, \zeta^r) - [c\left((k^s, u^s), (\xi^r, \zeta^r)\right) + \lambda \mathcal{R}(T)]\right) \quad (22)$$

is the c-transform of the physics-regularized Kantorovich dual problem. Then the optimization problem Eq. (13) becomes

$$\sup_f \left[ \int f(k^t, u^t) dP^t - \int f^{c,-}(k^s, u^s) dP^s \right], \quad (23)$$

which is exactly the physics-regularized Kantorovich problem.

Therefore, if $(T^*, f^*)$ is the saddle point solution of Eq. (13), then $f^*$ is an optimal solution of Eq. (23), $\mathcal{L}(P^s, T^*, f^*) = K(P^s, P^t) + \mathcal{R}(T^*)$, which verifies the first assertion of the theorem.

**(2)** The saddle point $(T^*, f^*)$ satisfy

$$T^*(k^s, u^s) \in argmax_{(\xi^r, \zeta^r)} f^*(\xi^r, \zeta^r) - [c\left((k^s, u^s), (\xi^r, \zeta^r)\right) + \lambda \mathcal{R}(T)] \ a.s. \quad (24)$$

$$\Longrightarrow f^{*c,-}(k^s, u^s) = f^*(T^*(k^s, u^s)) - [c\left((k^s, u^s), T^*(k^s, u^s)\right) + \lambda \mathcal{R}(T)] \quad (25)$$

where $f^{*c,-}(k^s, u^s) = \sup_{(\xi^r, \zeta^r)} \left(f^*(\xi^r, \zeta^r) - [c\left((k^s, u^s), (\xi^r, \zeta^r)\right) + \lambda \mathcal{R}(T)]\right)$.

With condition $T^*_\# P^s = P^t$, the pushforward distribution $P^r = P^t$, then from the construction of $\mathcal{R}(T)$, we have $\mathcal{R}(T) = 0$. Thus Eq. (13) degenerates to

$$\sup_f \inf_T \int_{\Omega_k \times \Omega_u} c\left((k, u), T(k, u)\right) - f(T(k, u)) dP^s + \int_{\Omega_k \times \Omega_u} f(k, u) dP^t, \quad (26)$$

which is exactly the dual form of the primal Monge problem. Then we have

$$
\begin{aligned}
&\int_{\Omega} c\left((k^s, u^s), T^*(k^s, u^s)\right) dP^s \\
&= \int_{\Omega} f^*(T^*(k^s, u^s)) dP^s - \int_{\Omega} f^{*c,-}(k^s, u^s) dP^s \\
&= \int_{\Omega} f^*(k^t, u^t) dP^t - \int_{\Omega} f^{*c,-}(k^s, u^s) dP^s \\
&= \int_{\Omega \times \Omega} f^*(k^t, u^t) - f^{*c,-}(k^s, u^s) d\pi \\
&\leqslant \int_{\Omega \times \Omega} c\left((k^s, u^s), (k^t, u^t)\right) d\pi, \ \forall \pi \in \Pi(P^s, P^t)
\end{aligned}
\tag{27}
$$

Take infimum on both sides of the inequation, we obtain

$$
\begin{aligned}
&\inf_T \int_{\Omega} c\left((k^s, u^s), T^*(k^s, u^s)\right) dP^s \\
&\leqslant \inf_\pi \int_{\Omega \times \Omega} c\left((k^s, u^s), (k^t, u^t)\right) d\pi \\
&\leqslant \int_{\Omega} c\left((k^s, u^s), T(k^s, u^s)\right) dP^s,
\end{aligned}
\tag{28}
$$

where $T$ is any map that satisfies $(Id, T)_{\#} P^s = \pi \in \Pi(P^s, P^t)$. Therefore, the solution of the Monge problem exists and $T^*$ is the Monge map. □

### B.3  Algorithm analysis

---
**Algorithm 1:** Optimization of POTT

---
1 **Input:** source data $\hat{\mathcal{D}}_s$, pretrained model $\mathcal{G}_\eta$, target data $\hat{\mathcal{D}}_t$;
2 Initialize $T_\theta, f_\phi$;
3 **for** $N_{11}$ *steps* **do**
4      Freeze $\phi$, update $\theta$ to minimize the first objective of Eq. (14) for $N_{12}$ steps;
5      Freeze $\theta$, update $\phi$ to maximize the first objective of Eq. (14) with $\theta$;
6 **end**
7 **for** $N_2$ *steps* **do**
8      Freeze $\theta$ and $\phi$, update $\eta$ to minimize the second objective in Eq. (14);
9 **end**
10 **Output:** $\mathcal{G}_\eta$ as approximation of $\mathcal{G}_t$.

---

Thus, the entire training process of POTT requires $O((N_{11} \cdot N_{12} + N_2)B(C_\phi + C_\eta + C_\theta))$ operations, where $C_\phi, C_\eta, C_\theta$ are model parameters, B is the batchsize, $N_{12}$ is typically set to 10. As the dataset size grows larger, $N_{11}$ and $N_2$ should be set larger, and the model size of $T_\theta$ and $f_\phi$ also grow. In summary, the computational efficiency of POTT is similar to other neural OT methods.

## C  Experiment details

### C.1  Simulation datasets

The relative Mean Square Error $rMSE = \|u_{pred} - u_{gt}\|_2^2 / \|u_{gt}\|_2^2$ is reported for evaluation, where $u_{gt}$ denotes the ground truth of output u.

Simulation datasets used for evaluation are generated as follows:

### C.1.1  Burgers' equation

Considering the 1-D Burgers' equation on unit torus:

$$
u_t + u u_x = \nu u_{xx}, \ x \in (0, 1), t \in (0, 1],
\tag{29}
$$

we aim to learn the operator mapping the initial condition to the solution funciton at time one, i.e. $\mathcal{G}_\theta : u_0 = u(x, 0) \to u(x, 1)$. As shown in Tab. 8, we differ the generation of $u_0$ and parameter $\nu$ to construct different domains:

Table 8: Generation of $u_0$ and parameter settings in 1-d Burgers' equation.

| SUB-DOMAIN | DESCRIPTION |
|---|---|
| $\mathcal{D}_1$ | $u_0 \sim \mathcal{N}(0, 7^2(-\Delta + 7^2 \mathcal{I})^{-2}), \nu = 0.01$ |
| $\mathcal{D}_2$ | $u_0 \sim \mathcal{N}(0.2, 49^2(-\Delta + 7^2 \mathcal{I})^{-2.5}), \nu = 0.002$ |
| $\mathcal{D}_3$ | $u_0 \sim \mathcal{N}(0.5, 625^2(-\Delta + 25^2 \mathcal{I})^{-2.5}), \nu = 0.004$ |

The $\mathcal{N}$ denotes the normal distribution. The resolution of x-axis is 1024.

We assess the domain shift between these domains in a rather straightforward way. Specifically, we regard the data-driven model trained with sufficient target data as **Oracle**, and compare the prediction error between the finetuning method trained with 100 target data and the Oracle. Tasks with lower gap are considered as simpler than tasks with larger gap. As shown in Tab. 9, the prediction gap in tasks $\mathcal{D}_2 \to \mathcal{D}_1$, $\mathcal{D}_2 \to \mathcal{D}_3$, $\mathcal{D}_3 \to \mathcal{D}_1$ are smaller than the other three, which means models are easier to transfer in these tasks. So we only conduct comparision experiment in tasks $\mathcal{D}_1 \to \mathcal{D}_2$, $\mathcal{D}_1 \to \mathcal{D}_3$ and $\mathcal{D}_3 \to \mathcal{D}_2$.

Table 9: Evaluation of tasks' difficulties of Burgers' equation.

| METHOD | $\mathcal{D}_1 \to \mathcal{D}_2$ | $\mathcal{D}_1 \to \mathcal{D}_3$ | $\mathcal{D}_2 \to \mathcal{D}_1$ | $\mathcal{D}_2 \to \mathcal{D}_3$ | $\mathcal{D}_3 \to \mathcal{D}_1$ | $\mathcal{D}_3 \to \mathcal{D}_2$ |
|---|---|---|---|---|---|---|
| FINETUNING | 0.1191 | 0.0801 | 0.0381 | 0.0786 | 0.0252 | 0.0938 |
| ORACLE | 0.0402 | 0.0403 | 0.0140 | 0.0403 | 0.0140 | 0.0402 |

As shown in Tab. 2, the improvement of POTT compared to finetuning is substantial. When the amount of target data is only 50, POTT reduced the relative error by 23.64% (from 0.2001 to 0.1528) in task $\mathcal{D}_1 \to \mathcal{D}_2$ and by 19.21% (from 0.1546 to 0.1249) in task $\mathcal{D}_3 \to \mathcal{D}_2$. When the amount of target data is 100, although the performance of finetuning greatly improves, POTT still largely reduces the relative error by 18.98% and 19.30% in task $\mathcal{D}_1 \to \mathcal{D}_2$ and $\mathcal{D}_3 \to \mathcal{D}_2$. In the relatively simple task $\mathcal{D}_1 \to \mathcal{D}_3$, although finetuning already achieves satisfactory results, POTT can still reduce the relative error by about 10%, while TL-DeepONet, DARE-GRAM and COD even caused negative transfer and result in larger relative error.

### C.1.2 Darcy flow

The 2-d Darcy flow takes the form
$$\nabla \cdot (k(x)\nabla u(x)) = 1, \ x \in [0, 1] \times [0, 1]$$
$$u(x) = 0, \ x \in \partial([0, 1] \times [0, 1]). \tag{30}$$

We aim to learn the operator mapping the diffusion coefficient $k(x)$ to the solution function $u(x)$, i.e. $\mathcal{G}_\theta : k(x) \to u(x)$. We use the leading 100 terms in a truncated $Karhunen - Lo\grave{e}ve$ (KL) expansion for a Gaussian process with zero mean and covariance kernel $\mathcal{K}(x)$ to generate $a(x)$, and construct different function domains by differ the kernel $\mathcal{K}(x, x')$, as shown in Tab. 10.

The $\Omega_{square}$ denotes a square with vertice on $\{(0, 0), (0, 1), (1, 0), (1, 1)\}$ in $[0, 1] \times [0, 1]$, $\Omega_{triangle}$ denotes a triangle with vertice on $\{(0, 0), (0, 1), (0.5, 1)\}$ in $[0, 1] \times [0, 1]$. The resolution of $x \in [0, 1] \times [0, 1]$ is $64 \times 64$. Similarly, the difficulties of transfer tasks are shown in Tab. 11. As discussed in Sec. 5, we conduct experiments in the hard tasks $\mathcal{D}_2 \to \mathcal{D}_1$, $\mathcal{D}_1 \to \mathcal{D}_3$ and $\mathcal{D}_2 \to \mathcal{D}_3$. Results are provided in Tab. 3.

### C.1.3 Advection equation

The Advection equation takes the form
$$u_t + \nu u_x = 0, \ x \in (0, 1), t \in (0, 1]. \tag{31}$$

Table 10: Generation of $a(x)$ in Darcy flow.

| SUB-DOMAIN | DESCRIPTION |
|:---:|:---:|
| $\mathcal{D}_1$ | $\mathcal{K}(x, x') = exp(-\frac{\|x-x'\|_2^2}{2}), \Omega_u = \Omega_{square}$ |
| $\mathcal{D}_2$ | $\mathcal{K}(x, x') = exp(-\frac{\|x-x'\|_2^2}{2}), \Omega_u = \Omega_{triangle}$ |
| $\mathcal{D}_3$ | $\mathcal{K}(x, x') = exp(-\frac{\|x-x'\|_1^2}{2}), \Omega_u = \Omega_{square}$ |

Table 11: Evaluation of tasks' difficulties of Darcy flow.

| METHOD | $\mathcal{D}_1 \to \mathcal{D}_2$ | $\mathcal{D}_1 \to \mathcal{D}_3$ | $\mathcal{D}_2 \to \mathcal{D}_1$ | $\mathcal{D}_2 \to \mathcal{D}_3$ | $\mathcal{D}_3 \to \mathcal{D}_1$ | $\mathcal{D}_3 \to \mathcal{D}_2$ |
|:---|:---:|:---:|:---:|:---:|:---:|:---:|
| FINETUNING | 0.0429 | 0.1605 | 0.0869 | 0.3553 | 0.0469 | 0.0598 |
| ORACLE | 0.0111 | 0.0682 | 0.0153 | 0.0682 | 0.0153 | 0.0111 |

We aim to learn the operator mapping the initial condition to the solution funciton at a continuous time set $[0, 1]$, i.e. $\mathcal{G}_\theta : u_0 = u(x, 0) \to u(x, t)$. As shown in Tab. 12, we differ the function types and generation process of $u_0$ and parameter $\nu$ to construct different domains:

Table 12: Generation of $u_0$ and parameter settings in Advection equation.

| SUB-DOMAIN | DESCRIPTION |
|:---:|:---:|
| $\mathcal{D}_1$ | $u_0(x) = ax^2 + bx + c, \quad a, b, c \in \mathcal{U}(-1, 1), \nu = 3$ |
| $\mathcal{D}_2$ | $u_0(x) = ax^3 + bx^2 + cx + d, \quad a \in \mathcal{U}(0, 1), b, c \in \mathcal{U}(-0.5, 0.5), d = 0.5, \nu = 2$ |
| $\mathcal{D}_3$ | $u_0(x) = a sin(bx + c), \quad a \in \mathcal{U}(0, 1), b \in \mathcal{U}(5, 10), c \in \mathcal{U}(-1, 1), \nu = 1$ |

The $\mathcal{U}$ denotes the uniform distribution. The resolution of x-axis and t-axis are 100 and 50, respectively. Similarly, the difficulties of transfer tasks are shown in Tab. 13. For this equation, we select three easier tasks to explore the effectiveness of POTT in easy tasks, i.e., the $\mathcal{D}_1 \to \mathcal{D}_2$, $\mathcal{D}_2 \to \mathcal{D}_1$ and $\mathcal{D}_3 \to \mathcal{D}_2$. As shown in Tab. 4, although the performance of finetuning is satisfying, POTT can further reduce the prediction error to a lower level.

## C.2 Real-world datasets

We use the preprocessed version of ERA5 from WeatherBench [29] as real-world datasets for climate forecasting task. We utilize the ClimODE [33] as backbone model and conduct experiment based on their settings and codes, which are all available from their paper. Note that we only report the latitude-weighted RMSE for evaluation, since the Anomaly Correlation Coefficient (ACC) reported in [33] is rather closed for most methods in comparision. The definition of the latitude-weighted RMSE is

$$\text{RMSE} = \frac{1}{N} \sum_t^N \sqrt{\frac{1}{HW} \sum_h^H \sum_w^W \alpha(h)(u_{gt} - u_{pred})^2},$$

where $\alpha(h) = \cos(h)/\frac{1}{H}\sum_{h'}^H \cos(h')$ is the latitude weight, $u_{gt}$ and $u_{pred}$ are ground truth and model prediciton respectively. Lower latitude-weighted RMSE values indicate better model performance in capturing spatial or climate patterns. Visualized results and analysis are provided in Sec. 5, details results are provided in Tab. 14.

## C.3 Implementation details

**Simulation experiment.** To test the generalizability of POTT with different models, we employed different backbones on various datasets. On the Burgers' equation dataset, $\mathcal{G}_\eta$ is parametrized as a 1-d Fourier Neural Operator (FNO) model, $T_\theta$ is an operator network composed of two fully connected networks (FCN), and $f_\phi$ is an FCN. On the Advection equation and Darcy flow datasets, $\mathcal{G}_\eta$ adopt

Table 13: Evaluation of tasks' difficulties of Advection equation.

| METHOD | $\mathcal{D}_1 \to \mathcal{D}_2$ | $\mathcal{D}_1 \to \mathcal{D}_3$ | $\mathcal{D}_2 \to \mathcal{D}_1$ | $\mathcal{D}_2 \to \mathcal{D}_3$ | $\mathcal{D}_3 \to \mathcal{D}_1$ | $\mathcal{D}_3 \to \mathcal{D}_2$ |
|---|---|---|---|---|---|---|
| FINETUNING | 0.0143 | 0.0795 | 0.0891 | 0.0777 | 0.2226 | 0.0723 |
| ORACLE | 0.0123 | 0.0134 | 0.0252 | 0.0134 | 0.0252 | 0.0153 |

Table 14: Latitude weighted RMSE($\downarrow$) results of global forecasting on ERA5 dataset. Oracle denotes the model trained with sufficient target data.

| | Lead-Time (hours) | Source-only | ClimODE Finetuning | POTT | Oracle | ClimaX Oracle |
|---|---|---|---|---|---|---|
| z | 6 | $265.71 \pm 14.76$ | $216.77 \pm 14.58$ | $196.53 \pm 14.89$ | $102.9 \pm 9.3$ | 247.5 |
| | 12 | $405.83 \pm 27.34$ | $295.89 \pm 22.92$ | $251.4 \pm 24.93$ | $134.8 \pm 162.3$ | 265.3 |
| | 18 | $568.39 \pm 35.93$ | $381.6 \pm 29.34$ | $325.7 \pm 32.28$ | $12.7 \pm 14.4$ | 319.8 |
| | 24 | $723.61 \pm 43.37$ | $445.45 \pm 35.35$ | $374.68 \pm 39.66$ | $193.4 \pm 16.3$ | 364.9 |
| | 36 | $1127.46 \pm 58.67$ | $598.61 \pm 44.72$ | $506.43 \pm 51.64$ | $259.6 \pm 22.3$ | 455.0 |
| t | 6 | $1.81 \pm 0.07$ | $1.63 \pm 0.07$ | $1.36 \pm 0.07$ | $1.16 \pm 0.06$ | 1.64 |
| | 12 | $2.72 \pm 0.14$ | $2.17 \pm 0.09$ | $1.85 \pm 0.09$ | $1.32 \pm 0.13$ | 1.77 |
| | 18 | $3.51 \pm 0.19$ | $2.5 \pm 0.1$ | $1.89 \pm 0.11$ | $1.47 \pm 0.16$ | 1.93 |
| | 24 | $4.51 \pm 0.26$ | $2.71 \pm 0.11$ | $1.91 \pm 0.12$ | $1.55 \pm 0.18$ | 2.17 |
| | 36 | $6.96 \pm 0.42$ | $3.08 \pm 0.14$ | $2.54 \pm 0.19$ | $1.75 \pm 0.26$ | 2.49 |
| t2m | 6 | $9.09 \pm 0.21$ | $3.43 \pm 0.24$ | $2.65 \pm 0.2$ | $1.21 \pm 0.09$ | 2.02 |
| | 12 | $9.83 \pm 0.24$ | $4.09 \pm 0.25$ | $3.1 \pm 0.21$ | $1.45 \pm 0.10$ | 2.26 |
| | 18 | $9.97 \pm 0.21$ | $3.66 \pm 0.14$ | $2.89 \pm 0.2$ | $1.43 \pm 0.09$ | 2.45 |
| | 24 | $9.69 \pm 0.36$ | $3.01 \pm 0.17$ | $2.01 \pm 0.31$ | $1.40 \pm 0.09$ | 2.37 |
| | 36 | $10.91 \pm 0.67$ | $3.86 \pm 0.27$ | $3.13 \pm 0.28$ | $1.70 \pm 0.15$ | 2.87 |
| u10 | 6 | $2.22 \pm 0.11$ | $2.19 \pm 0.11$ | $2.01 \pm 0.11$ | $1.41 \pm 0.07$ | 1.58 |
| | 12 | $3.09 \pm 0.15$ | $2.71 \pm 0.12$ | $2.42 \pm 0.13$ | $1.81 \pm 0.09$ | 1.96 |
| | 18 | $3.8 \pm 0.16$ | $3.11 \pm 0.13$ | $2.6 \pm 0.13$ | $1.97 \pm 0.11$ | 2.24 |
| | 24 | $4.48 \pm 0.17$ | $3.36 \pm 0.13$ | $2.75 \pm 0.14$ | $2.01 \pm 0.10$ | 2.49 |
| | 36 | $6 \pm 0.21$ | $4.05 \pm 0.15$ | $3.24 \pm 0.17$ | $2.25 \pm 0.18$ | 2.98 |
| v10 | 6 | $2.45 \pm 0.15$ | $2.34 \pm 0.13$ | $2.15 \pm 0.14$ | $1.53 \pm 0.08$ | 1.60 |
| | 12 | $3.4 \pm 0.22$ | $2.88 \pm 0.14$ | $2.6 \pm 0.16$ | $1.81 \pm 0.12$ | 1.97 |
| | 18 | $4.03 \pm 0.26$ | $3.15 \pm 0.14$ | $2.73 \pm 0.17$ | $1.96 \pm 0.16$ | 2.26 |
| | 24 | $4.58 \pm 0.25$ | $3.44 \pm 0.15$ | $2.86 \pm 0.19$ | $2.04 \pm 0.10$ | 2.48 |
| | 36 | $5.72 \pm 0.25$ | $4.21 \pm 0.18$ | $3.36 \pm 0.22$ | $2.29 \pm 0.24$ | 2.98 |

a 2-d DeepONet model, $T_\theta$ has a structure similar to $\mathcal{G}_\eta$, and $f_\phi$ is a convolutional neural network (CNN). We use Adam as optimizer and the learning rate is $1e - 3$ for all tasks. The learning rate of the backbone of $\mathcal{G}_\eta$ is ten times smaller than the last two layers, which is a widely-used technique in transfer learning. A cosine annealing strategy is adopted for learning rate of $\mathcal{G}_\eta$. Details of model architectures and data generation can be found in codes provided by [22, 25].

**Climate forecasting.** The architecture and the training details of the data-driven model $\mathcal{G}_\eta$ are consistent with the codes provided by [33]. The architecture and training procedures of $T_\theta$ and $f_\phi$ are the same with that in the simulation experiment.

All experiments are conducted on a single 16GB NVIDIA 4080 device.

# D    Further discussion

## D.1    Practical value of POTT

As the absolute errors of POTT are still too large for practical applications, we discuss the practical value of POTT compared to finetuning with more target data.

- POTT is proposed as a model-agnostic transfer learning framework designed to improve performance under limited supervision. As is standard in transfer learning literature, our primary focus is on the relative gain over baselines such as source-only and finetuning, rather

| $\mathcal{D}_1 \rightarrow \mathcal{D}_3$ | 200 | 400 | 600 | 800 | 1000 |
|---|---|---|---|---|---|
| FT | 0.1482 | 0.1391 | 0.1001 | 0.0928 | 0.0867 |
| POTT | 0.1309 | 0.1195 | 0.0899 | 0.0817 | 0.0792 |

Table 15: Results in the extreme few-shot settings.

| $\mathcal{D}_2 \rightarrow \mathcal{D}_3$ | 200 | 400 | 600 | 800 | 1000 |
|---|---|---|---|---|---|
| FT | 0.2339 | 0.1668 | 0.1267 | 0.1055 | 0.0947 |
| POTT | 0.2086 | 0.1487 | 0.1007 | 0.0914 | 0.0834 |

Table 16: Results in the extreme few-shot settings.

than the absolute error, which also depends on the model's expressiveness. In our experiments, all methods use the same architecture to ensure a fair comparison.

- Similar evaluation protocols are widely adopted in transfer learning research. For example, in domain adaptation tasks on DomainNet using ResNet-101, SOTA methods achieve only about 40% absolute accuracy. Nevertheless, they are considered effective due to their consistent improvement over source-only baselines (about 25%). This demonstrates that the relative improvement is a well-accepted metric for evaluating transfer methods.

- It seems that the errors of about 0.15 and 0.21 in Darcy flow with 50 and 100 target samples are still too large for practical applications. However, this is mainly due to the limited capacity of the baseline model. Even with 2000 labeled samples in domain $\mathcal{D}_3$, the fully supervised model reaches an error of about 0.07. This highlights the inherent difficulty of the task. To investigate further, we conducted scaling experiments as shown in Tab. 15 and Tab. 16. The results confirm that while increasing the amount of labeled data does improve performance, POTT can achieve comparable accuracy with significantly fewer samples. For instance, POTT with 600 800 samples matches finetuning with 1000 samples, which is very close to the oracle performance achieved by the backbone model ( 0.07).

- In many practical applications such as high-resolution simulations, chaotic weather systems, or medical diagnostics, collecting a moderate number of labeled samples is feasible, but obtaining large-scale annotations remains prohibitively expensive. In such scenarios, finetuning alone may fall short, and incorporating principled transfer methods like POTT can effectively compensate for the data scarcity.

In summary, while achieving very low absolute error (e.g., <0.05) may require more powerful models or larger datasets, our results clearly show that POTT significantly improves performance in the low-data regime, reducing the required number of labeled target samples to achieve similar accuracy as finetuning. This highlights its practical value in scenarios where labeled data is limited but not extremely scarce — a common situation in many scientific and engineering domains.

### D.2 Relations between the reported evaluation metric and the residuals.

Whlie the evaluation metric used in our experiments is the relative $L_2$ error, it is closely related to the residuals of the governing equations, which is widely used in PDEs learning literature. In fact, it is equivalent to evaluating the residuals of the equations.

For the simulation datasets (e.g., Burgers, Darcy), the ground-truth solutions are generated by high-accuracy numerical solvers, and thus satisfy the PDEs with negligible error. For the advection equation, the ground truth is given by its analytical solution. Given this, the error $du = u_{gt} - u_{pred}$ reflects the deviation from the true solution. By the differential mean value theorem, we have

$$F(u_{pred}) = F(u_{gt}) + F'(u^*)(u_{pred} - u_{gt}) = F'(u^*)du, \tag{32}$$

for some $u^*$ between $u_{gt}$ and $u_{pred}$. Here, $F$ is the differential operator defined by the PDE. This shows that the residual $F(u_{pred})$ is proportional to the error $du$, with the proportionality factor being the derivative of the operator at some intermediate point. Therefore, our reported prediction errors provide an equivalent proxy for assessing whether the transferred model respects the underlying physics.

### D.3 Training time of POTT

The main computational overhead of POTT lies in training the transport map $T$, but this cost is both amortizable and reducible.

**Amortization:** The cost of training $T$ is relatively independent of the cost of training the downstream model $\mathcal{G}$. In scenarios where $\mathcal{G}$ is large or expensive to train—as is often the case with real-world scientific models—the rel-

| | Finetuning | COD | DAREGRAM | POTT |
|---|---|---|---|---|
| Darcy flow | 5min | 7min | 11min | 20min |
| ERA5 | 80min | - | - | 100min |

Table 17: Wall-clock training time comparision.

ative overhead of POTT becomes modest. We report approximate wall-clock training times on Darcy flow and ERA5 datasets in Tab. 17. As shown, POTT introduces a moderate additional cost, which is acceptable given the substantial improvements in generalization. On simpler tasks such as Darcy, where training $\mathcal{G}$ is very fast, the overhead is more visible, but still within practical limits.

**Reducibility:** As noted in the limitations section, a promising direction for reducing the training cost of POTT is to leverage large pre-trained models from related domains. For example, incorporating pre-trained multimodal foundation models, and applying parameter-efficient tuning or distillation techniques to train the POTT map, can significantly reduce the overhead while potentially improving performance.

### D.4 How sensitive are results to the choice of metric and to the accuracy of the surrogate operator ?

**Choice of metric $m$:** We use the metric $m$ in Eq. (12) to quantify the distributional discrepancy between $P_u^r$ and $\hat{\mathcal{G}}(P_k^r)$. In principle, $m$ can be any nonparametric divergence or distance measure, such as the Wasserstein distance. In our implementation, we adopt the L2 metric for its simplicity and computational efficiency. Empirically, we observed that L2 provides stable gradients and effective regularization. Preliminary experiments with alternative choices (e.g., L1) yield comparable trends, suggesting robustness to the choice of metric. Nevertheless, systematically studying the impact of different metrics is an important direction for future work.

**Accuracy of surrogate operator $\hat{\mathcal{G}}$:** Following standard transfer learning assumptions, we rely on a well-trained source model $\hat{\mathcal{G}}$ that captures the physics in the source domain. If $\hat{\mathcal{G}}$ is poorly trained, it may introduce misleading inductive biases. To mitigate this, we ensure that $\hat{\mathcal{G}}$ is trained with sufficient labeled data and validated carefully before transfer. In practice, we also monitor the influence of the marginal-consistency term during training and observe that its contribution is stable, suggesting that the learned physics from the source is indeed beneficial under the domain shift.

