# OpenReview forum: "A Physics-preserved Transfer Learning Method for Differential Equations"
_NeurIPS.cc/2025/Conference — NeurIPS 2025 poster_

### Official Review · Reviewer_6yj9 · 2025-06-16

**Clarity:** 2
**Significance:** 2
**Originality:** 3
**Rating:** 3
**Confidence:** 3

**Summary:**

This paper studied the domain shift issue in the learning of neural operators of PDE problems, proposed a transfer learning methods which can preserve the physical relation of the equation during training.
This is acomplished by the optimal tensor transport method and a physical regularization.
Three simulation bechmarks  and a real-world climate forcasting datasets were applied to validate the effectiveness of the method.

**Questions:**

1. The authors state that POTT is more valuable when the amount of target sample is small, yet not extremely small. However, when target sample is small, the prediction methods are too large for all the methods in the paper. When traget sample increase, simple finetuning can have good prediction on target domain. So what's the advantage of this method in practice?

2. Can the authors show the computational time comparison of POTT method with other baseline methods?

3. How about the physical regularization term $\mathcal{R}(T)$ compared to the physics-informed loss like in [1]? $\mathcal{R}(T)$  is problem-depended, making it hard to extend to arbitrary PDE problems.  On the contrary, the physics-informed loss is a universal framework once the equations are known.

[1] Wang S, Wang H, Perdikaris P. Learning the solution operator of parametric partial differential equations with physics-informed DeepONets[J]. Science advances, 2021, 7(40): eabi8605.

**Ethical Concerns:**

["NO or VERY MINOR ethics concerns only"]

**Final Justification:**

The primary reason I am unable to recommend acceptance of this paper is the limited extent of the improvement demonstrated. The chosen error metric, rMSE, is essentially the square of the more commonly used relative L2 error, which diminishes the apparent performance gains. Additionally, the manuscript contains several noticeable typographical errors.

**Limitations:**

yes

**Quality:**

3

**Strengths And Weaknesses:**

Strengths:

The paper present a detailed analysis of existing transfer learning methods for PDE problems.
They  propose the Physics-preserved Optimal Tensor Transport (POTT) method to learn a physics-preserved optimal transport map between source and target domains.
In the case of small training samples on target domain, the POTT method can largely improve the generalization ability of the model.
The method is fully evaluated and analyzed.

Weakness:

The Optimal Tensor Transport method is not well-presented for readers not familiar with OT theory, causing the difficulty of understanding the motivation of this work. The practical POTT loss (14) is not well-described. The physical regularization term $\mathcal{R}(T)$ is problem-depended, without a universal framework like PINN loss, making the method hard to extend to arbitrary PDE problems.

---

> ### Author Rebuttal · Authors · 2025-07-30
>
> Thank you for taking the time to review our paper and provide professional and constructive reviews. We are encouraged by the positive comments on the full evaluation and analysis. All suggestions will be carefully incorporated into final version.
>
> **_Q: Comparison between R(T) and physics-informed loss._**
>
> Thank you for raising this insightful question.
>
> We respectfully argue that the proposed physical regularization term R(T) is **more general and broadly applicable** than the physics-informed loss. Our reasoning is as follows:
>
> -- **Physics-informed loss requires fully known equations, which is a strong and often impractical assumption.** In many real-world applications, only partial physical knowledge is available (e.g., certain necessary conditions), making such methods inapplicable.
>
> -- **R(T) can leverage partial physical priors.** When partial physical knowledge (e.g., conservation properties) is available, R(T) can be constructed accordingly, e.g., **line 194**. In such cases, the physics-informed loss is unusable, whereas our method still applies. In fact, when full equation forms are available, R(T) can recover the same behavior as physics-informed loss—thus subsuming them as **a special case.**
>
> -- **R(T) also support the general settings with no explicit prior available.** As described in **line 191–199**, R(T) can be defined without any explicit physical priors by regularizing marginal distribution by the source model $G^s$.
>
> In both our synthetic and real-world experiments, we intentionally do **not** provide the explicit equations during training or inference. Despite this, R(T) is still effective, while physics-informed losses cannot be applied.
>
> Therefore, while physics-informed losses are indeed powerful when the equations are known, we believe R(T) offers a more flexible and universal framework that is better suited to the realities of many scientific and engineering applications.
>
> **_Q: The advantage of POTT in practice._**
>
> Thank you for raising this practical concern.
> We agree that when the target domain has a large number of labeled samples, simple finetuning methods can achieve good performance. However, in many real-world applications, obtaining labeled data in the target domain is costly, time-consuming, or even infeasible—especially in scientific and industrial domains such as marine environmental forecasting, climate forecasting, and medical imaging, etc.
>
> POTT is specifically designed for scenarios where the target sample size is limited but not extremely small. In this medium-data regime, standard finetuning may suffer from underfitting due to insufficient supervision, while methods assuming access to a large target dataset are inapplicable. POTT leverages source-domain information in a more principled and physics-aware way to improve generalization, offering a clear performance advantage in such regimes.
>
> In short, POTT is particularly valuable **in settings where label acquisition is expensive**, in which  data is scarce, or governed by complex processes.
>
> **_Q: The computational time comparison of POTT method with other baseline methods._**
>
> The main computational overhead of POTT lies in training the transport map $T$. However, this cost is both amortizable and reducible.
>
> **Amortization**: The cost of training $T$ is relatively independent of the cost of training the downstream model $G$. Therefore, in tasks where training $G$ is computationally expensive, the relative overhead of training $T$ becomes negligible. To illustrate this, we report the approximate training times on two datasets:
>
> |      | Finetuning | COD | DAREGRAM | POTT |
> |------|----------|------|------------|----------|
> | Darcy | 5min     | 7min | 11min | 20min |
> | ERA5 | 80min    | - | - | 100min|
>
> On simpler tasks like Darcy, where $G$ is small and FT is fast, POTT's additional cost is more noticeable. However, in more complex tasks like ERA5, where training $G$ is inherently expensive, the overhead introduced by POTT becomes much less significant in proportion. In many real-world applications where labeled data is scarce or costly, this trade-off—using additional computation to gain improved generalization—can be beneficial.
>
> **Reducibility**: As noted in the limitations section, a promising direction for reducing the training cost of POTT is to leverage large pre-trained models from related domains. For example, incorporating pre-trained multimodal foundation models, and applying parameter-efficient tuning or distillation techniques to train the POTT map, can significantly reduce the overhead while potentially improving performance.

---

> > ### Comment · Reviewer_6yj9 · 2025-08-02
> >
> > Thanks for the detailed responses. Despite the improvement of POTT over finetuning on medium target data size, I still have concerns on the practical issue "POTT is specifically designed for scenarios where the target sample size is limited but not extremely small". In Table 3, the average (not 'ADVERGE') error is 0.21 for $n^t=50$ and 0.15 for $n^t=100$, and an error increase of 0.05 for finetuning. The errors are both too large for practical applications.  I think an error below 0.05 on target domain is acceptable(since the error on source domain may less than 0.001). Can it only be achieved by adding more target data, rather than by improving the method?

---

> > > ### Author Response · Authors · 2025-08-05
> > >
> > > Thank you for the thoughtful comment and for raising a key practical concern. We address this from four perspectives:
> > >
> > > 1.	POTT is proposed as a model-agnostic transfer learning framework designed to improve performance under limited supervision. As is standard in transfer learning literature, **our primary focus is on the relative gain over baselines such as source-only and finetuning, rather than the absolute error, which also depends on the model’s expressiveness.** In our experiments, all methods use the same architecture to ensure a fair comparison.
> > >
> > > 2.	Similar evaluation protocols are **widely adopted in transfer learning research**. For example, in domain adaptation tasks on DomainNet using ResNet-101, SOTA methods [1, 2] achieve only \~40% absolute accuracy. Nevertheless, they are considered effective due to their consistent improvement over source-only baselines (~25%). This demonstrates that **the relative improvement is a well-accepted metric for evaluating transfer methods**.
> > >
> > > 3.	We agree that errors of $0.15–0.21$ (for $n^t=50,100$) may be too large for applications. However, this is mainly due to the limited capacity of the baseline model. Even with **2000 labeled samples** in domain $D_3$, the fully supervised model reaches an error of **\~0.07**. This highlights the inherent difficulty of the task. To investigate further, we conducted scaling experiments as shown below:
> > >
> > > |$D_2\to D_3$ |  200  |  400  |  600  |  800  | 1000  |
> > > |------------|-------|-------|--------|-------|------|
> > > |FT   	    | 0.2339 |  0.1668 | 0.1267 | 0.1055 | **0.0947** |
> > > |POTT	    | 0.2086 |  0.1487 | **0.1007** | **0.0914** | 0.0834 |
> > >
> > > $D_1\to D_3$ |  200  |  400  |  600  |  800 |  1000 |
> > > |------------|-------|-------|-------|------|-------|
> > > |FT   	    | 0.1482 | 0.1391 | 0.1001 | 0.0928 | **0.0867** |
> > > |POTT	    | 0.1309 | 0.1195 | **0.0899** | **0.0817** | 0.0792 |
> > >
> > > These results confirm that **while increasing the amount of labeled data does improve performance, POTT can achieve comparable accuracy with significantly fewer samples.** For instance, POTT with **600\~800 samples** matches finetuning with **1000 samples**, which is very close to the oracle performance achieved by the backbone model (**\~0.07**).
> > >
> > > 4.	In many practical applications such as high-resolution simulations, chaotic weather systems, or medical diagnostics, collecting a moderate number of labeled samples is feasible, but obtaining large-scale annotations remains prohibitively expensive. In such scenarios, **finetuning alone may fall short**, and **incorporating principled transfer methods like POTT can effectively compensate for the data scarcity.**
> > >
> > > **In summary**, while achieving very low absolute error (e.g., <0.05) may require more powerful models or larger datasets, our results clearly show that POTT significantly improves performance in the low-data regime, **reducing the required number of labeled target samples to achieve similar accuracy as finetuning.** This highlights its practical value **in scenarios where labeled data is limited but not extremely scarce — a common situation in many scientific and engineering domains.**
> > >
> > > If you have any other questions, please let us know and we will do our best to answer. Thank you.
> > >
> > > [1] Ren et al. H3T: Hierarchical Transferable Transformer with TokenMix for Unsupervised Domain Adaptation. Expert Systems with Applications, 2025.
> > >
> > > [2] Wang et al. Probability-Polarized Optimal Transport for Unsupervised Domain Adaptation. AAAI, 2024.

---

### Official Review · Reviewer_kQhX · 2025-06-22

**Clarity:** 1
**Significance:** 2
**Originality:** 2
**Rating:** 4
**Confidence:** 2

**Summary:**

The paper presents a transfer learning algorithm for operator learning problems based on optimal transport. By adding a physics-preserving regularization term to the transport problem, physical properties of the system should be preserved in the transfer. Experiments on both simulation, as well as weather/climate data show performance improvements over classical finetuning, as well as domain adaptation methods.

**Questions:**

- Could you provide a scaling experiment for the amount of target data needed for the method to work (better than finetuning)?
- Could you provide an experiment with a more complicated PDE, e.g. incompressible Navier-Stokes? There it is also possible to check whether incompressibility under domain shift is retained.
- For all experiments, could you provide baselines that are trained from scratch on target data (with the same number of samples as used for POTT)?
- To check whether physics was preserved during the transfer procedure, could you provide e.g. the residuals of the equations?

**Ethical Concerns:**

["NO or VERY MINOR ethics concerns only"]

**Final Justification:**

The paper seems a good contribution, although the results are not too strong. The main reason for my score is the language and clarity. It has to be improved in a CRV.

**Limitations:**

yes

**Paper Formatting Concerns:**

no concerns.

**Quality:**

2

**Strengths And Weaknesses:**

# Strengths

The paper presents a novel transfer learning framework based on optimal transport (POTT) and tailored to operator learning problems in physics. It is tested on both simulation, as well as experimental/real-world data.

# Weaknesses

The paper suffers from numerous language issues that hinder readability -- e.g., should the title not be "A Physics-preserving Transfer Learning Method for Differential Equations"? There are also several typos in the manuscript. Improving the language would enhance the accessibility of the content.

The framework still needs a certain amount of samples from the target distribution to work (as also outlined in the limitations). A scaling experiment would be interesting to see.

Experiments on the simulation data only consider simple equations of which only one is nonlinear. The robustness of the framework under more complicated datasets is not fully shown.

With the given experiments and baselines, it is not clear whether using POTT over training from scratch on target data is favorable.

The sub-domains used in the experiments with simulation data are nowhere defined in the main text, hindering understanding; please give a short explanation in the main text.

Finally, as the title suggests, the framework should preserve physics. From the experiments it is not entirely clear to me whether this is the case. A visual analysis (as on the last page) does not seem sufficient.

---

> ### Author Rebuttal · Authors · 2025-07-31
>
> Thank you for taking the time to review our paper and provide professional and constructive reviews. We are encouraged by the positive comments on the novelty of POTT, and the thoroughness of our experimental evaluation. We will carefully incorporate all suggestions into the final version of the paper.
>
> **_Q: Scale experiment of the amount of target data._**
>
> Thank you for raising this important question. In extreme few-shot settings (e.g., ≤10 target samples), both finetuning and POTT indeed face significant challenges due to the risk of overfitting. **From a distribution-matching perspective, fitting a meaningful target distribution with only a few samples is fundamentally ill-posed** and prone to instability. This limitation applies broadly to all distribution-based methods.
>
> To empirically evaluate this, we conduct a scale-down experiment on the Darcy flow task $\mathcal{D}_2 \to \mathcal{D}_3$ under various numbers of labeled target samples $n^t$:
>
> |  $n^t$   |  100  |  50   |  20   |  10   |  5   |   1  |
> |----------|-------|-------|-------|-------|-------|------|
> | Tgt. Only  | 0.3493 | 0.3925 | 0.4130 | **0.4897** | **0.5107** | **0.5568** |
> | Finetuning | 0.3553 | 0.4693 | 0.4866	 | 0.5368 | 0.5521 | 0.6282 |
> | POTT     | **0.2271** | **0.3527** | **0.4126** | 0.5257 | 0.5364 | 0.6359 |
>
> We observe that:
>
> --As $n^t$ decreases, all methods degrade in performance.
>
> --When $n^t >= 20$, POTT still provides advantages over both **finetuning** and **training from scratch**.
>
> --When $n^t <= 10$, both POTT and finetuning begin to overfit, and performance starts to deteriorate significantly.
>
> --At the extreme setting $n^t = 1$, POTT fails entirely, as the transport map becomes unreliable and may even hurt the learning of the target predictor due to **inaccurate distribution matching**.
>
> These results clarify that POTT is most effective in the moderate few-shot regime, but may not be applicable when labeled target data is extremely scarce. That said, our main focus is on scenarios where labeled data is limited but not vanishingly small, which aligns with many real-world cases in scientific machine learning where acquiring a small number of high-quality labels is feasible.
>
> We also believe that incorporating robust priors, meta-learning, or foundation models can potentially extend POTT's applicability to extreme few-shot regimes, and we consider this a meaningful direction for future work.
>
> **_Q: Experiment with more complicated PDEs._**
>
> Thank you for the insightful suggestion.
> We acknowledge the importance of evaluating the method on more complex PDEs. While the synthetic equations in our main experiments—**Burgers’ equation and Darcy flow**—are **nonlinear and widely used** in the operator learning literature, we agree that more complicated dynamics can further demonstrate the robustness of POTT.
>
> To this end, we provide a new experiment on the incompressible Navier–Stokes (NS) equation, a canonical example of nonlinear, multivariate, and stiff dynamics. The 2D NS equation reads:
>
> $$
> \begin{align}
> \partial_t w(x, t) + u(x, t) \cdot \nabla w(x, t) &= \nu \Delta w(x, t) + f(x), &\quad x \in (0, 1)^2, \, t \in (0, T] \\\\
> \nabla \cdot u(x, t) &= 0, &\quad x \in (0, 1)^2, \, t \in [0, T]
> \end{align}
> $$
>
> where 𝑢=(𝑢,𝑣) is the velocity field, p is the pressure, and ν is the viscosity coefficient. We simulate domain shifts by varying ν∈{1e−3, 1e−4, 1e−5 }, denoted as D1, D2, D3, respectively. The settings of dataset are identical with the settings in [1]. Due to the rebuttal time constraint, we provide partial but representative results here:
>
> |    NS    | $\mathcal{D}_1 \to \mathcal{D}_2$ | $\mathcal{D}_1 \to \mathcal{D}_3$ |
> |$n^t$     |   50  |  100	 |  50  |  100  |
> |----------|-------|-------|------|-------|
> | Tgt. Only  | 0.2716 | 0.2368 | 0.3307 | 0.2793 |
> | Finetuning | 0.1903 | 0.1478 | 0.3188 | 0.2679 |
> | POTT     | **0.1749** | **0.1297** | **0.2706** | **0.2458** |
>
> We observe that in this more challenging scenario, model performance degrades significantly under low-data regimes. The target-only and finetuning baselines struggle to generalize, especially when $n^t=50$. In contrast, POTT is able to leverage the pretrained source model and capture structural information more effectively, leading to consistent improvements.
>
> This additional experiment supports the conclusion that POTT is robust to domain shifts even in highly nonlinear and multivariate PDEs like Navier–Stokes. We will include the full experimental setup and results in the final version of the paper.
>
> **_Q: Experiments that trained from scratch on target data._**
>
> Thank you for the suggestion. We agree that including target-only baselines trained from scratch helps clarify the effectiveness of POTT. Below, we provide comprehensive comparisons across all tasks.
>
> | Bur.  |  $D_1 \to D_2$ |  $D_1 \to D_3$ | $D_3 \to D_2$ |
> | $n^t$ |  50   |  100  |  50  |  100  |  50  |  100  |
> |------|-------|-------|------|-------|-------|-------|
> | Tgt. only | 0.2142 | 0.1429 | 0.1173 | 0.0968 | 0.2142 | 0.1429 |
> | Finetuning | 0.2001 | 0.1191 | 0.1049 | 0.0801 | 0.1546 | 0.0938 |
> | POTT | 0.1528 | 0.0965 | 0.0950 | 0.0705 | 0.1249 | 0.0757 |
>
> | Adv.  |  $D_1 \to D_2$ |  $D_2 \to D_1$ | $D_3 \to D_2$ |
> | $n^t$ |  50   |  100  |  50  |  100  |  50  |  100  |
> |------|-------|-------|------|-------|-------|-------|
> | Tgt. only  | 0.2162 | 0.1261 | 0.2585 | 0.1382 | 0.2162 | 0.1261 |
> | Finetuning | 0.0247 | 0.0143 | 0.2193 | 0.0891 | 0.1257 | 0.0723 |
> | POTT | 0.0207 | 0.0112 | 0.1872 | 0.0787 | 0.1016 | 0.0613 |
>
> | Darcy |  $D_2 \to D_1$ |  $D_1 \to D_3$ | $D_2 \to D_3$ |
> | $n^t$ |  50   |  100  |  50  |  100  |  50  |  100  |
> |------|-------|-------|------|-------|-------|-------|
> | Tgt. only | 0.1815 | 0.1122 | 0.3925 | 0.2893 | 0.3925 | 0.2893 |
> | Finetuning | 0.1426 | 0.0896 | 0.1556 | 0.1605 | 0.4693 | 0.3553 |
> | POTT | 0.1362 | 0.0762 | 0.1397 | 0.1404 | 0.3527 | 0.2271 |
>
> Across all tasks, POTT consistently outperforms both training from scratch and finetuning, demonstrating its effectiveness.
>
> **_Q: Residuals of the equations._**
>
> Thank you for the suggestion. The results we report ($|| \hat u - u||^2$, i.e., the MSE between the predicted solution $\hat u$ and the ground truth solution $u$) are effectively **equivalent to evaluating the residuals of the equations**.
>
> For the simulation datasets (e.g., Burgers, Darcy), the ground-truth solutions are generated by high-accuracy numerical solvers, and thus satisfy the PDEs with negligible error. For the advection equation, the ground truth is given by its analytical solution. Given this, the error **$du = u - \hat u$** reflects the deviation from the true solution.
>
> By the differential mean value theorem, we have
> $$F(\hat u) = F(u) + F'(v)du = F'(v)du, $$
> for some $v$ between $\hat u$ and $u$. This suggests that **when $|| \hat u - u||^2 = (du)^2$ (the reported error) is small, the residual $F(\hat u)$ is also small.** Therefore, our reported prediction errors provide an equivalent proxy for assessing whether the transferred model respects the underlying physics.
>
> **_Q: Typos._**
>
> Thank you for pointing this out.
>
> Regarding the title: while "Physics-preserving" is indeed more commonly used, our usage of "Physics-preserved" was intentional. Specifically, we aim to emphasize that **the physics, either learned by the source model or explicitly available from the problem, are preserved during the transfer process under the guidance of the POTT map $T$.** In this sense, "physics-preserved" refers to the outcome of the POTT method, rather than the transfer process itself being a physics-preserving mechanism. We acknowledge that this nuance may not be immediately clear to readers, and we will consider revising the title to improve clarity and alignment with standard phrasing.
>
> As for the rest of the manuscript, we will carefully proofread the paper to correct any remaining typos and improve the overall language quality for better readability and accessibility.
>
> **_Q: Definition of the sub-domains._**
>
> Thank you for the suggestion.
>
> In the main text, Tab. 1 provides visual examples of samples from all sub-domains used in the simulation experiments. We chose to include these visualizations because **they offer a more intuitive and accessible understanding of the domain gaps between sub-domains**, which we believe is more helpful than formal definitions for most readers.
>
> Besides, **the precise definitions of all sub-domains are provided in Appendix C.1**, and we explicitly refer to this in **the caption of Tab. 1** to ensure that interested readers can easily locate the relevant details.
>
> We appreciate your suggestion and will consider including a brief summary of the sub-domain definitions directly in the main text or Tab. 1 in the final version to improve clarity.
>
> [1] Li, Zongyi, et al. "Fourier neural operator for parametric partial differential equations." ICLR, 2021.

---

> > ### Comment · Reviewer_kQhX · 2025-08-06
> >
> > Thank you for the detailed rebuttal and answering my points. I think that including these points in the paper will improve it. I will therefore raise my score.

---

> > > ### Author Response · Authors · 2025-08-07
> > >
> > > Thank you for your positive feedback and for raising the score. Your comments are greatly appreciated and will be carefully incorporated in the final revision of our paper.

---

### Official Review · Reviewer_DNte · 2025-06-27

**Clarity:** 2
**Significance:** 3
**Originality:** 3
**Rating:** 5
**Confidence:** 3

**Summary:**

The paper formulates domain shift in data-driven PDE solvers as the joint presence of distribution bias and operator bias.  To address these difficulties, the authors propose Physics-preserved Optimal Tensor Transport (POTT): an optimal-transport map between source and target product distributions that is regularised to respect available physical laws.  The push-forward of the source data through the learnt map augments the small target set, after which the pretrained source operator is fine-tuned.  A dual formulation with a Lagrange multiplier network enables end-to-end min–max optimisation.  Experiments on three simulated PDE families (Burgers, Advection, Darcy) and a cross-region ERA5 climate forecast show consistent error reductions over finetuning and recent domain-adaptation baselines.

**Questions:**

1. For cases without explicit priors you rely on a marginal-consistency penalty (Eq. 12).  How sensitive are results to the choice of metric *m* and to the accuracy of the surrogate operator \( \hat G_s \)?
2.  What is the wall-clock training time of POTT relative to vanilla finetuning on the Darcy 64 × 64 task and on ERA5?
3. Can POTT handle ≤ 10 target samples?  Does the OT map overfit in that regime?
4. Have you evaluated against diffusion-based neural OT (e.g., Daniels et al., 2021) or Monge-Map networks trained without physics loss?
5.  Does the tensor-product OT map scale cubically with spatial dimension?  Any plans for hierarchical or low-rank variants?

**Ethical Concerns:**

["NO or VERY MINOR ethics concerns only"]

**Final Justification:**

I believe the main contribution lies in the idea of "physics preservation." After the rebuttal phase and reviewing the discussion between the authors and other reviewers, I gained a clearer understanding of the authors’ argument. And the  additional experiments show the "P" do have practical gain. I raised my score to accept.

**Limitations:**

Yes

**Quality:**

3

**Strengths And Weaknesses:**

## Strengths
1.  The distinction between distribution bias and operator bias offers a neat lens through which to view transfer learning for neural operators.
2.  Embedding physical constraints directly into an OT objective is intuitive and, to my knowledge, new in the context of operator transfer.  The regulariser design is flexible (explicit priors vs. marginal consistency).
3.   Six transfer tasks per equation plus a real-world ERA5 case demonstrate applicability to 1-D, 2-D and spatio-temporal settings.  Relative-error gains over finetuning reach 25–36 % on the hardest Darcy task.
4.   Removing the physics regulariser (OTT baseline) degrades qualitative fidelity, confirming its utility.
5.   Architecture details, optimisation schedules and hardware are documented; code release is promised.

##  Weaknesses


1. Thm 4.2 proves a consistency property of the dual saddle point but does **not** bound the transfer error nor quantify how much physics is preserved.
2. POTT introduces two additional networks and a bi-level training loop; training cost vs. plain finetuning is not reported.
3.  Comparisons omit recently proposed neural-OT approaches and physics-aware finetuning techniques; climate study omits domain-adaptation baselines for fairness reasons.
4. The method is said to “diminish when only a handful of target samples are available,” but no quantitative exploration of shot regimes is given.
5.  Several checklist items remain “\[TODO\]”; some typos (e.g., “foruc on” page 3).  Condensing Sections 3–4 would improve readability.

---

---

> ### Author Rebuttal · Authors · 2025-07-31
>
> Thank you for taking the time to review our paper and provide professional and constructive reviews. We are encouraged by the positive comments on our problem formulation, the novelty of POTT, and the thoroughness of our experimental evaluation. We will carefully incorporate all suggestions into the final version of the paper.
>
> **_Q: Thm 4.2 proves a consistency property of the dual saddle point but does not bound the transfer error nor quantify how much physics is preserved._**
>
> Thank you for the insightful comment. Thm. 4.2 focuses on justifying the consistency of the dual formulation and improving the conceptual understanding of POTT. It does not provide an explicit transfer error bound nor quantify the extent to which physics is preserved. **We agree that developing such theoretical guarantees is a very important and promising direction for future work.**
>
> As an **initial idea**, we can consider adapting the generalization bounds from semi-supervised domain adaptation. A typical result can be written as:
> $$
> \varepsilon^t \leq \varepsilon^s + \varepsilon^{lt} + \text{dist}(\mathcal{D}^s, \mathcal{D}^t),
> $$
> which controls the target error using source error, supervised error on labeled target samples, and a distribution discrepancy term that typically relies on the feature distribution alignment.
>
> However, as discussed in **Sec. 3.2 (lines 150–166)**, POTT does not rely on feature alignment but instead directly corrects operator bias via target domain characterization. Thus, a coarse decomposition for POTT might take the form:
>
> $$\varepsilon^t \leq \varepsilon^{lt} + \varepsilon^{r} + \text{dist}(\mathcal{D}^t, \mathcal{D}^{lt} \cup \mathcal{D}^{r}),$$
>
> where $\varepsilon^{lt}$ and $\varepsilon^{r}$ denote the supervised loss on labeled target data and pushforward samples respectively, and the distributional term $\text{dist}(\mathcal{D}^t, \mathcal{D}^{lt} \cup \mathcal{D}^{r})$ reflects **how well the target domain is characterized by POTT**. Further decomposing this term, especially under the physics regularization $R(T)$, requires integrating optimal transport theory with physics-preserved learning — a challenging but promising direction we plan to explore.
>
> As for quantifying how much physics is preserved, a simple idea is to measure this indirectly through prediction accuracy on physics-governed variables (e.g., $u$ in Darcy flow), which reflects compliance with the underlying PDE. While this provides a practical proxy, we acknowledge that developing more explicit metrics would give a more rigorous evaluation of physics preservation.
>
> We thank the reviewer again for raising this important point. Establishing theoretical bounds on transfer error and physics preservation will be a key focus of our future work.
>
> **_Q: How sensitive are results to the choice of metric m and to the accuracy of the surrogate operator $\hat G^s$ ?_**
>
> Metric m: We use the metric m in Eq. (12) to quantify the distributional discrepancy between $P^r_u$ and $\hat G^s(P^r_k)$. In principle, **m can be any nonparametric divergence or distance measure, such as the Wasserstein distance.** In our implementation, we adopt the L2 metric for its simplicity and computational efficiency. Empirically, we observed that L2 provides stable gradients and effective regularization. Preliminary experiments with alternative choices (e.g., L1) yield comparable trends, suggesting robustness to the choice of metric. Nevertheless, systematically studying the impact of different metrics is an important direction for future work.
>
> Accuracy of surrogate operator $\hat G^s$: **Following standard transfer learning assumptions, we rely on a well-trained source model $\hat G^s$ that captures the physics in the source domain.** If $\hat G^s$ is poorly trained, it may introduce misleading inductive biases. To mitigate this, we ensure that $\hat G^s$ is trained with sufficient labeled data and validated carefully before transfer. In practice, we also monitor the influence of the marginal-consistency term during training and observe that its contribution is stable, suggesting that the learned physics from the source is indeed beneficial under the domain shift.
>
> **_Q: Training time of  POTT._**
>
> The main computational overhead of POTT lies in training the transport map T, but this cost is both **amortizable** and **reducible**.
>
> **Amortization**: The cost of training T is relatively independent of the cost of training the downstream model G. In scenarios where G is large or expensive to train—as is often the case with real-world scientific models—the relative overhead of POTT becomes modest. We report approximate wall-clock training times on two representative datasets:
> |      | Finetuning | COD | DAREGRAM | POTT |
> |------|----------|------|------------|----------|
> | Darcy | 5min     | 7min | 11min | 20min |
> | ERA5 | 80min    | - | - | 100min|
>
> As shown, POTT introduces a moderate additional cost, which is acceptable given the substantial improvements in generalization. On simpler tasks such as Darcy, where training G is very fast, the overhead is more visible, but still within practical limits.
>
> **Reducibility**: As noted in the limitations section, a promising direction for reducing the training cost of POTT is to leverage large pre-trained models from related domains. For example, incorporating pre-trained multimodal foundation models, and applying parameter-efficient tuning or distillation techniques to train the POTT map, can significantly reduce the overhead while potentially improving performance.
>
> **_Q: Comparison against diffusion-based neural OT or vanilla Monge-Map networks._**
>
> Thank you for the insightful question. POTT is a general transfer learning framework and **does not impose restrictions on the underlying model used to learn the transport map T**. This flexibility means that more advanced techniques—such as diffusion-based neural OT methods—can in principle **be incorporated as backbones for learning the POTT map.** Moreover, using such methods or leveraging pre-trained diffusion-based OT models could improve both the quality and efficiency of POTT, which we highlighted as a promising direction in our **limitations section.**
>
> Regarding vanilla Monge-map networks trained without physics loss, we partially evaluated this in our paper: **in Fig. 5, we include results from standard OT-based transfer (labeled “OTT”)**, which fits transport solely via distribution alignment without any physics constraints. As discussed in **lines 215–222**, if a true Monge map exists and the network manages to learn it exactly, distribution alignment alone would suffice. However, in practice—especially under complex, high-dimensional settings—such ideal conditions rarely hold. Without physics guidance, the learned mapping might introduce significant bias or inconsistency with the target physics, leading to poor generalization. This highlights the necessity of our physics-aware training objective. Here we provide numerical results on task consistent with that in Fig. 5:
>
> |Finetuning | POTT w/o R | POTT  |
> |-------------|--------------|---------------|
> |0.0575    | 0.0543     | **0.0496** |
>
> **_Q: Extreme few-shot settings._**
>
> Thank you for raising this important question. In extreme few-shot settings (e.g., ≤10 target samples), both finetuning and POTT indeed face significant challenges due to the risk of overfitting. **From a distribution-matching perspective, fitting a meaningful target distribution with only a few samples is fundamentally ill-posed** and prone to instability. This limitation applies broadly to all distribution-based methods.
>
> To empirically evaluate this, we conduct a scale-down experiment on the Darcy flow task $\mathcal{D}_2 \to \mathcal{D}_3$ under various numbers of labeled target samples $n^t$:
>
> |  $n^t$   |  100  |  50   |  20   |  10   |  5   |   1  |
> |----------|-------|-------|-------|-------|-------|------|
> | Tgt. Only  | 0.3493 | 0.3925 | 0.4130 | **0.4897** | **0.5107** | **0.5568** |
> | Finetuning | 0.3553 | 0.4693 | 0.4866	 | 0.5368 | 0.5521 | 0.6282 |
> | POTT     | **0.2271** | **0.3527** | **0.4126** | 0.5257 | 0.5364 | 0.6359 |
>
> We observe that:
>
> -- As $n^t$ decreases, all methods degrade in performance.
>
> -- When $n^t >= 20$, POTT still provides advantages over both **finetuning** and **training from scratch**.
>
> -- When $n^t <= 10$, both POTT and finetuning begin to overfit, and performance starts to deteriorate significantly.
>
> -- At the extreme setting $n^t = 1$, POTT fails entirely, as the transport map becomes unreliable and may even hurt the learning of the target predictor due to **inaccurate distribution matching**.
>
> These results clarify that POTT is most effective in the moderate few-shot regime, but may not be applicable when labeled target data is extremely scarce, which aligns with many real-world cases in scientific machine learning where acquiring a small number of high-quality labels is feasible.
>
> We also believe that incorporating robust priors, meta-learning, or foundation models can potentially extend POTT's applicability to extreme few-shot regimes, and we consider this a meaningful direction for future work.
>
> **_Q: Computational effective variants._**
>
> Thank you for the valuable question. The computational complexity does increase with spatial dimensionality. However, whether it scales cubically depends on the choice of backbone model. We acknowledge that improving the scalability of POTT is an important direction for future work. Promising approaches include using efficient backbone architectures (e.g., hierarchical structure), proper training strategy (e.g., low-rank training) or subspace decompositions.
>
>
> **_Q: Typos._**
>
> We sincerely appreciate the reviewer's careful reading. All reported typos will be corrected in the final manuscript, and we will conduct a full proofread to ensure textual accuracy throughout the paper.

---

> > ### Comment · Reviewer_DNte · 2025-08-08
> >
> > I appreciate the author's effort for detailed explanation and more experiments. I have no further questions.

---

> > > ### Author Response · Authors · 2025-08-08
> > >
> > > Thank you very much for your positive feedback. Your comments are greatly appreciated and will be carefully incorporated into the final revision of our paper.

---

### Official Review · Reviewer_JnQq · 2025-06-30

**Clarity:** 3
**Significance:** 3
**Originality:** 3
**Rating:** 4
**Confidence:** 2

**Summary:**

This paper introduced a transfer learning method, POTT, to solve differential equations in the context of domain shifts. The authors mainly tackle two challenges: distribution bias and operator bias. By learning an optimal transport map between the source and target domains, constrained by physics-based regularization, the method ensures both domain adaptation and perservation of physical properties. The method was evaluated on both simulation and real-world datasets.

**Questions:**

1. Have you analyzed the trade-off between the computational cost of the method and the potential gains in performance? Are there any comparative experiments showing how the method performs relative to existing methods in terms of computational efficiency (more interesting on a large dataset)?

2. In cases where physical priors are unavailable, have you considered alternative formulations, maybe learned priors, which can better capture complex physical relationships? Did you explore this direction?

3. Several typos: line 50 "foruc" -> "focus", line 268 target smaples -> target samples; Table 3: ADVERGE -> AVERAGE

4. Some figures are missing color bars, for example, Figures 1, 4, 5. And can you explain how to compare those pictures across different methods?

5. Is PINN a comparable method to POTT? I noticed that no PINN-based methods were included in the comparisons.

**Ethical Concerns:**

["NO or VERY MINOR ethics concerns only"]

**Final Justification:**

The paper presents a novel method for solving differential equations under domain shifts, and the results show that it outperforms existing methods. The reason for my score remaining as borderline accept is that the high computational cost remains a concern for practical applications, and the paper only demonstrates POTT on one real-world dataset.

**Limitations:**

yes

**Paper Formatting Concerns:**

There are no major formatting issues.

**Quality:**

3

**Strengths And Weaknesses:**

**Strength**

1. The paper is well-written, with a clear description of the problem and method.
2. The paper addressed an interesting and important problem in solving differential equations under domain shifts.
3. The results demonstrate that POTT outperforms existing methods.

**Weakness**
1. For the general cases with no physical priors available, the physical regularization term $R$ relies on marginal distributions. It is unclear if this will oversimplify the underlying physics, potentially influencing performance. Additionally, the paper lacks an ablation study to evaluate how this regularization term influences the results.

2. The evaluation mainly focused on synthetic datasets. Also, there is no discussion of the method's robustness to noisy or incomplete data, which are common in real-world scenarios.

3. The paper mentioned that the method has a high computational cost due to its reliance on min-max optimization. This could pose challenges for scaling the method to high-dimensional DEs or more complex real-world problems.

---

> ### Author Rebuttal · Authors · 2025-07-30
>
> Thank you for taking the time to review our paper and provide professional and constructive reviews. We are encouraged by the positive comments on the significance of the addressed problem and the paper writing. All suggestions will be carefully incorporated into final version.
>
> **_Q: Physical regularization term in general cases with no physical priors available._**
>
> Thank you for this insightful question.
>
> As discussed in **lines 191–199**, we distinguish between two scenarios regarding the availability of physical priors when constructing the physical regularization term R. When explicit physical priors are available, we leverage them directly to define R. In more general cases where no explicit priors are accessible, we instead derive R based on the marginal of the pushforward distribution.
> First, it is worth emphasizing that **physical priors are often more accessible than they might initially appear**. In our formulation, physical priors are not limited to the exact equations (sufficient conditions); they also include weaker but still informative necessary conditions that characterize certain properties of the system. For example, in the climate forecasting experiment, while we lack a complete physical model due to the extreme complexity of the problem, we can still rely on basic conservation principles to formulate necessary constraints. These partial insights enable the construction of R as shown in Eq. (11).
>
> Second, in cases where no physical priors are explicitly available, POTT constructs R with marginal distributions of \(P^r\) . Importantly, this should not be viewed as a naive statistical constraint. As shown in Eq. (12), the regularization is guided by \(G_s\) learned from the source domain, which inherently captures physical properties shared across domains. This mechanism can be interpreted as **a form of learned prior**—one that encodes structural knowledge implicitly learned from the source domain. Given that source and target domains often share common physical principles, this transfer serves as an effective proxy for physical priors in the absence of explicit ones.
>
> We acknowledge that such a learned prior may not capture all domain-specific physical nuances, particularly those unique to the target domain. However, it does retain the underlying physics shared across domains, which is often able to improve generalization.
>
> In summary, our definition of physical priors is broad and includes both sufficient and necessary conditions, making it applicable to a wide range of problems. Even in the most challenging cases without any accessible prior knowledge, POTT provides a practical and principled approach to incorporating learned priors.
>
> We appreciate the reviewer for raising this point—it has helped us better clarify and articulate the role and scope of the regularization term R. We will complete the discussion in Sec. 4.1
>
> **_Q: Ablation study for regularization term._**
>
> We've presented an ablation study on the physical regularization term R in **lines 290–299** and in **Fig. 5**. As illustrated, when the physical constraint R is removed, the model degenerates into standard OTT-based transfer. In this case, the generated sample (shown in the 4th subfigure) tend to violate physical property. In contrast, incorporating the generalized physical regularization based on Eq. (12), POTT yields transformed sample (3rd subfigure) that better preserve the underlying physical structure.
>
> To further highlight the impact of the physical regularization term, we have included additional ablation results on the same task as in Fig. 5. In this extended comparison, we isolate the effect of R by comparing models with and without it, under identical settings. The results consistently show that the absence of R leads to degraded model performance.
>
> | Finetuning | POTT w/o R | POTT |
> |------------|------------|-----------|
> | 0.0575     | 0.0543     | **0.0496** |
>
> This provides clear empirical evidence that R is essential for enforcing physical consistency even in scenarios where explicit physical priors are unavailable.
>
> **_Q: Trade-off between the computational cost and the potential gains._**
>
> The main computational overhead of POTT lies in training the transport map \(T\). However, this cost is both amortizable and reducible.
>
> Amortization: The cost of training \(T\) is relatively independent of the cost of training the downstream model \(G\). Therefore, in tasks where training \(G\) is computationally expensive, the relative overhead of training \(T\) becomes negligible. To illustrate this, we report the approximate training times on two datasets:
>
> |      	  | Darcy | ERA5|
> |------------|------------|------|
> |Finetuning | 5min  | 80min|
> |POTT     | 20min | 100min|
>
> On simpler tasks like Darcy, where G is small and FT is fast, POTT's additional cost is more noticeable. However, in more complex tasks like ERA5, where training G is inherently expensive, the overhead introduced by POTT becomes much less significant in proportion. In many real-world applications where labeled data is scarce or costly, this trade-off—using additional computation to gain improved generalization—can be beneficial.
>
> Reducibility: As noted in the limitations section, a promising direction for reducing the training cost of POTT is to leverage large pre-trained models from related domains. For example, incorporating pre-trained multimodal foundation models, and applying parameter-efficient tuning or distillation techniques to train the POTT map, can significantly reduce the overhead while potentially improving performance.
>
> **_Q: Datasets for Evaluation._**
>
> Although our synthetic experiments are based on known physical equations for data generation, **they are carefully designed to reflect realistic scenarios. ** Specifically, we sample functions from diverse distributions such that each input-output pair may correspond to a different latent equation. Crucially, **no equation-specific information (e.g., symbolic expressions or coefficients) is provided during training. ** This setup mimics real-world conditions where governing equations are unknown or variable. Further details are provided in the appendix.
>
> In addition to synthetic datasets, we also evaluate POTT on a real-world dataset—ERA5 for weather forecasting. ERA5 is based on historical climate records and naturally includes common real-world data issues. As shown in Fig. 3, POTT outperforms baseline methods under these challenging conditions.
>
> **_Q: Typos._**
>
> We sincerely appreciate the reviewer's careful reading. All reported typos will be corrected in the final manuscript, and we will conduct a full proofread to ensure textual accuracy throughout the paper.
>
> **_Q: Color bars for Fig. 1, 4, 5_**
>
> Thank you for pointing this out.
> While we mention in the figure captions that **brighter colors (yellow) indicate higher values**, we agree that including explicit color bars would make the visualizations clearer. **We will add them to Fig. 1, 4, and 5 in the final version of the paper.**
>
> Regarding cross-method comparison: the color mapping is consistent across Fig. 1, 4, and 5, where the color at each spatial location represents the function value—yellow indicates high values and dark blue indicates low values. The spatial distribution of colors reflects the structural properties of the function being visualized.
>
> Take Fig. 4 as an example. Col. 2 shows the ground truth, while Col. 3–5 show the predictions from POTT, FT, and COD, respectively. By visually comparing the color and shape patterns, one can observe that POTT’s predictions (Col. 3) more closely match the ground truth in both value and spatial structure, indicating better prediction accuracy and physics preservation. The same principle applies to other figures as well.
>
> **_Q: Relation with PINN-based methods._**
>
> While both PINN-based methods and our proposed POTT emphasize the integration of physics into the learning process, **they are not directly comparable. ** As described in **line 22–25** of the introduction, PINNs rely on the explicit form of the equations to define their physics-informed loss. Consequently, **they become inapplicable when the underlying equations are unknown or only partially specified. **
>
> In real-world tasks, we rarely have access to fully known equations. Even when an equation is available, factors such as uncertain coefficients or perturbed functional forms can significantly impair the generalization ability of PINNs. This limitation motivates the greater adoption of data-driven methods.
>
> Our experimental setup mimics this realistic scenario. Although our synthetic experiments are based on known equations for data generation, we deliberately **do not provide any equation information (e.g., symbolic expression or parameter values) ** during training. Besides, we sample function pairs drawn from function distributions, meaning each training example may correspond to a different latent equation. Under these conditions—without explicit equation knowledge—**PINNs cannot be trained, and therefore are not suitable comparison baselines.**

---

> > ### Comment · Reviewer_JnQq · 2025-08-04
> >
> > Thanks for your detailed responses! I do not have further questions.

---

> > > ### Author Response · Authors · 2025-08-06
> > >
> > > Thank you very much for your positive and timely feedback. Your comments are greatly appreciated and will be carefully reflected in the final revision of our paper.

---

### Note · Authors · 2025-08-12

We sincerely thank all reviewers and the AC for your effort and constructive feedback, which have been invaluable for improving this work. Here we briefly summarize the rebuttal:

**_Strengths acknowledged by reviewers:_**

-- Clear formulation of the problem, introducing an intuitive and novel method.

-- Comprehensive evaluation on synthetic and real-world datasets, covering 1-D, 2-D, and spatio-temporal settings.

-- Clear, well-organized presentation.

**_Main concerns raised by reviewers:_**

-- **Methodology:** Definition and generality of the physical regularization term $R(T)$; lack of further theoretical analysis; relation to PINN-based methods.

-- **Experiments:** Training cost; performance in very limited target samples (1–5-shot); absence of target-only baselines; equation residuals; need for more complex PDE tests; discussion on absolute error reduction of POTT and its advantages over simply adding target data.

**_Rebuttal highlights:_**

**Methodology:**

-- Clarified the definition and mechanism of $R(T)$ with and without physics priors.

-- Outlined initial ideas for futher theoretical analysis.

-- Explained connections and differences with PINN-based methods, emphasizing POTT's generality beyond physics-informed losses.

**Experiments:**

-- Added training time comparisons, showing costs are amortizable and reducible.

-- Conducted scale experiments on few-shot settings, identifying POTT's effective low-data regime.

-- Included target-only baselines for all tasks.

-- Added experiments on the Navier–Stokes equation.

-- Demonstrated equivalence between reported metrics and equation residuals.

-- Clarified that POTT is a model-agnostic transfer learning method, where relative improvement is a more appropriate measure than absolute error.

-- Conducted scaling experiments, showing that while increasing the amount of labeled data does improve performance, POTT can achieve comparable accuracy with significantly fewer samples.

**Presentation:** Corrected typos and clarified figure details.

We are pleased that **most reviewers indicated their main concerns were resolved and reached a consensus on positive attitude toward this work.** We appreciate the reviewers’ recognition of the paper’s clear formulation, novel methodology, and comprehensive evaluation, and look forward to the opportunity for this work to contribute to the community.

---

### Decision · Program_Chairs · 2025-09-17

**Decision:**

Accept (poster)

**Comment:**

This paper introduces the Physics-preserved Optimal Tensor Transport (POTT) method for transfer learning in PDE problems. The contribution is novel and relevant, addressing data-scarce scenarios where labeled target data are costly or difficult to obtain. The method shows improved generalization over finetuning, supported by extensive evaluation. In their rebuttal, the authors clarify that the proposed physical regularization term R(T) is more flexible than physics-informed losses, as it can incorporate partial priors or even operate without explicit equations. They also argue that the computational overhead is modest in large-scale tasks and could be further reduced using pre-trained models.

However, concerns remain. The exposition of Optimal Tensor Transport is not easily accessible, and the description of the POTT loss is vague. More critically, the reported absolute error levels are still too high for practical applications, raising questions about utility. Overall, this paper is borderline: it offers novelty but limited practical impact.